# The rise and fall of the *Phytophthora infestans* lineage that triggered the Irish potato famine

Kentaro Yoshida[1†], Verena J Schuenemann[2†], Liliana M Cano[1], Marina Pais[1], Bagdevi Mishra[3,4,5], Rahul Sharma[3,4,5], Chirsta Lanz[6], Frank N Martin[7], Sophien Kamoun[1‡], Johannes Krause[2‡], Marco Thines[3,4,5,8‡], Detlef Weigel[9‡], Hernán A Burbano[9*]

[1]The Sainsbury Laboratory, Norwich, United Kingdom; [2]Institute of Archaeological Sciences, University of Tübingen, Tübingen, Germany; [3]Biodiversity and Climate Research Centre, Frankfurt, Germany; [4]Institute of Ecology, Evolution and Diversity, Goethe University, Frankfurt, Germany; [5]Senckenberg Gesellschaft für Naturforschung, Frankfurt, Germany; [6]Genome Center, Max Planck Institute for Developmental Biology, Tübingen, Germany; [7]Agriculture Research Services, United States Department of Agriculture, Salinas, United States; [8]Centre for Integrated Fungal Research, Frankfurt, Germany; [9]Department of Molecular Biology, Max Planck Institute for Developmental Biology, Tübingen, Germany

**Abstract** *Phytophthora infestans*, the cause of potato late blight, is infamous for having triggered the Irish Great Famine in the 1840s. Until the late 1970s, *P. infestans* diversity outside of its Mexican center of origin was low, and one scenario held that a single strain, US-1, had dominated the global population for 150 years; this was later challenged based on DNA analysis of historical herbarium specimens. We have compared the genomes of 11 herbarium and 15 modern strains. We conclude that the 19th century epidemic was caused by a unique genotype, HERB-1, that persisted for over 50 years. HERB-1 is distinct from all examined modern strains, but it is a close relative of US-1, which replaced it outside of Mexico in the 20th century. We propose that HERB-1 and US-1 emerged from a metapopulation that was established in the early 1800s outside of the species' center of diversity.

*For correspondence: hernan.burbano@tuebingen.mpg.de

†These authors contributed equally to this work

‡These authors are listed in alphabetical order

## Introduction

Potato late blight's impact on humankind is rivaled by few other plant diseases. The Spanish introduced Europeans to the South American staple crop potato shortly after their conquest of the New World, but for three centuries Europe stayed free of *P. infestans*, the causal agent of late blight. In 1845, the oomycete *P. infestans* finally reached Europe, spreading rapidly from Belgium to other countries of mainland Europe and then to Great Britain and Ireland. The impact of the epidemic reached catastrophic levels in Ireland, where the population was more dependent on potato for their subsistence than in other parts of Europe (*Bourke, 1964*; *Reader, 2009*). The subsequent Great Famine killed around 1 million people, and an additional million were forced to leave the island (*Turner, 2005*). Even today, the Irish population remains less than three quarters of what it was at the beginning of the 1840s. These dramatic consequences of the *P. infestans* epidemic were due to the absence of chemical and genetic methods to combat it; such means became available only several decades later.

Ever since triggering the Irish famine, *P. infestans* has continued to wreak havoc on potato fields throughout the world. Late blight remains the most destructive disease of the third largest food crop, resulting in annual losses of potatoes that would be sufficient to feed anywhere from 80 to many

**eLife digest** Few crop failures have been as devastating as those caused by potato late blight in the 1840s. This disease is caused by a filamentous microbe called *Phytophthora infestans,* which spread from North America to Europe in 1845, leading to the Great Famine in Ireland and to severe crop losses in the rest of Europe. *Phytophthora* is thought to have originated in the Toluca valley of Mexico, where many different strains evolve alongside wild potato relatives, but the exact strain that caused the Great Famine, and how it is related to modern strains of the pathogen, has remained a mystery.

Yoshida et al. have used a technique call 'shotgun' sequencing to map the genomes of 11 historical strains of *P. infestans* and 15 modern strains. The historical strains were extracted from the leaves of potato and tomato plants that were collected in North America and Europe, including Ireland and Great Britain, from 1845 onwards and stored in herbaria for future research. By comparing the genomes of the historical and modern samples, Yoshida et al. found that the historical strains all belonged to a single lineage that shows very little genetic diversity.

Previously it has been proposed that this lineage was the same as US-1, which was the dominant strain of potato blight in the world until the end of the 1970s, or that it was more closely related to modern strains than to US-1. Yoshida et al. now rule out both of these possibilities and show that the lineage that caused the great famine, which they call HERB-1, is clearly distinct from US-1, although they are closely related, and they conclude that both HERB-1 and US-1 might have dispersed from a common ancestor that existed outside of Mexico in the early 1800s. Why US-1 later replaced HERB-1 as the dominant strain in the world is an important question for future studies.

hundreds of millions of people (*Fisher et al., 2012*). *Phytophthora infestans* is an extraordinarily virulent and adaptable pathogen (*Fry, 2008*; *Haas et al., 2009*). In agricultural systems, sexual reproduction may trigger explosive population shifts that are driven by the emergence and migration of asexual lineages (*Fry et al., 1992*, *2009*; *Cooke et al., 2012*). The species is thought to originate from Toluca Valley, Mexico, where it infects wild relatives of potato, frequently undergoes sexual reproduction and co-occurs with the two closely related species *P. mirabilis* and *P. ipomoeae* (*Tooley et al., 1985*; *Goodwin et al., 1994*; *Flier et al., 2003*; *Grünwald and Flier, 2005*). In its center of origin, *P. infestans* is characterized by high levels of genetic and phenotypic diversity (*Grünwald and Flier, 2005*).

The genomes of a few *P. infestans* strains have been described (*Haas et al., 2009*; *Raffaele et al., 2010a*; *Cooke et al., 2012*). Compared to other species in the genus, the 240 Mb T30-4 reference genome of *P. infestans* is large, with three quarters of the genome consisting of repetitive DNA. A large number of genes codes for effector proteins, many of which are delivered inside plant cells to promote host colonization, for instance by suppressing plant immunity. RXLR proteins, the main class of host-translocated effectors, are encoded by about 550 genes in the *P. infestans* T30-4 genome. RXLR effectors that can be recognized by plant immune receptors, known as Resistance (R) proteins, are said to have 'avirulence' activity. Upon introduction of a cognate *R* gene into the host population, such avirulence effectors become a liability for the pathogen, and natural selection favors the spread of pseudogenized or mutated alleles (*Vleeshouwers et al., 2011*).

The detailed descriptions and drawings of Heinrich Anton de Bary and the reports of several other pioneers of plant pathology leave no doubt that the 19th century blight epidemic was triggered by *P. infestans* (*de Bary, 1876*; *Bourke, 1964*). What remains controversial is the relationship of the 19th century strains to modern isolates. The quest for understanding the origin of the strain that resulted in the Irish famine began with extant samples. Prior to the late 1970s, global *P. infestans* populations outside of South America and Mexico, the centers of diversity of the host and the pathogen, were dominated by a single clonal lineage that had the mitochondrial (mtDNA) haplotype Ib and was called US-1 (*Goodwin et al., 1994*). It was therefore proposed that the US-1 lineage was a direct descendant of the strain that first caused widespread late blight in North America from 1843 on, and then triggered the Irish famine beginning in 1845 (*Bourke, 1964*; *Goodwin et al., 1994*). This hypothesis was subsequently directly addressed by PCR analysis of infected 19th century potato leaves stored in herbaria. The conclusion from these studies was that the historic strains belonged to a mtDNA haplotype, Ia, that was distinct from that of the US-1 lineage (*Ristaino et al., 2001*; *May and Ristaino, 2004*).

Because Ia was at the time not only the predominant haplotype in the Toluca Valley in Mexico (*Gavino and Fry, 2002*; *Flier et al., 2003*), but had also been found in South America (*Perez et al., 2001*), *May and Ristaino (2004)* speculated that the 19th century and US-1 lineages represented two independent epidemics of divergent lineages that had both originated in South America and spread from there to North America and Europe. A caveat was that these far-reaching conclusions were based on only three mtDNA SNPs (*Ristaino et al., 2001*; *May and Ristaino, 2004*).

Since these first herbarium analyses, the retrieval and sequencing of DNA from museum specimens, fossil remains and archaeological samples—collectively known as ancient DNA (aDNA) (*Pääbo et al., 2004*)—have seen impressive advances thanks to the advent of high-throughput sequencing technologies. The combined analysis of modern and ancient genomes of human pathogens has begun to solve important questions about their history and evolution (*Bos et al., 2011*, *2012*). Here we show that aDNA methods hold similar promise for plant pathology and that they can improve our understanding of historically important plant pathogen epidemics.

To determine how the historic *P. infestans* strain(s) relate to extant isolates, we shotgun-sequenced 11 herbarium samples of infected potato and tomato leaves collected from continental Europe, Great Britain, Ireland, and North America in the period from 1845 to 1896, and extracted information on *P. infestans* mitochondrial and nuclear genomes. To understand the subsequent evolution of the pathogen, we compared the historic *P. infestans* genomes to those of 15 modern 20th century strains that span the genetic diversity of the species, and to the two sister species *P. ipomoeae* and *P. mirabilis*. Our analyses revealed that the 19th century epidemic was caused by a single genotype, HERB-1, that persisted for at least 50 years. While it is distinct from all examined modern strains, HERB-1 is closely related to the 20th century US-1 genotype, suggesting that these two pandemic genotypes may have emerged from a secondary metapopulation rather than from the species' Mexican center of diversity.

## Results

### Preservation of ancient DNA and genome statistics

Nineteenth-century samples of potato and tomato leaves with *P. infestans* lesions were obtained from the herbaria of the Botanische Staatssammlung München and the Kew Royal Botanical Gardens (*Table 1* and *Figure 1*). DNA was extracted under clean room conditions and two genomic libraries were prepared from each sample for Illumina sequencing. The preparations were expected to comprise *P. infestans* DNA, host DNA from potato or tomato as well as DNA from microbes that had colonized either the living material at the time of its collection, or the dried material during its storage in the herbaria.

The first set of libraries was used for verification of aDNA-like characteristics, and the second set was used for production. In this second set we used a repair protocol that removes aDNA-associated lesions, mainly characterized by cytosine deamination to uracil (U), which is subsequently converted to and read as thymine (T) (*Hofreiter et al., 2001*; *Briggs et al., 2007*, *2010*; *Brotherton et al., 2007*). The combination of uracil-DNA-glycosylase (UDG) and endonuclease VIII, which removes uracil residues and repairs abasic sites, reduces the overall per-base error rate to as low as one 20th of unrepaired aDNA (*Briggs et al., 2010*).

Ancient DNA fragments are typically shorter than 100 bp (*Pääbo, 1989*), and paired-end reads of 100 bases each will therefore substantially overlap. Forward and reverse reads from the unrepaired libraries (*Table 2*) were merged, requiring at least 11 base overlap. Merging of short-insert libraries considerably decreases the error-rate and also generates sequences that reflect the original molecule length (*Kircher, 2012*). The median length of merged reads was in the range of approximately 50–85 bp (*Figure 2A,B*).

Merged sequences were mapped to the *P. infestans* T30-4 reference genome (*Haas et al., 2009*). Deamination of C to U in aDNA is highest at the first base (*Briggs et al., 2007*), and C-to-T was the predominant substitution at the 5'-end of molecules (*Figure 2C,D*). Based on mapping against the reference genome, we estimated the fraction of *P. infestans* DNA in the samples to be between 1% and 20% (*Figure 2E*). Most of the remaining reads could be mapped to the reference genomes for potato and tomato (*Potato Genome Sequencing Consortium, 2011*; *The Tomato Genome Consortium, 2012*).

In addition to 11 historic samples from Ireland, Great Britain, continental Europe and North America (*Figure 1* and *Table 1*), we shotgun sequenced 14 modern strains from Europe, the Americas and

**Table 1.** Provenance of *P. infestans* samples

| | ID | Country of origin | Collection year | Host species | Reference* |
|---|---|---|---|---|---|
| Herbarium samples | KM177500 | England | 1845 | *Solanum tuberosum* | 1 |
| | KM177513 | Ireland | 1846 | *Solanum tuberosum* | 1 |
| | KM177502 | England | 1846 | *Solanum tuberosum* | 1 |
| | KM177497 | England | 1846 | *Solanum tuberosum* | 1 |
| | KM177514 | Ireland | 1847 | *Solanum tuberosum* | 1 |
| | KM177548 | England | 1847 | *Solanum tuberosum* | 1 |
| | KM177507 | England | 1856 | *Petunia hybrida* | 1 |
| | M-0182898 | Germany | Before 1863 | *Solanum tuberosum* | 2 |
| | KM177509 | England | 1865 | *Solanum tuberosum* | 1 |
| | M-0182900 | Germany | 1873 | *Solanum lycopersicum* | 2 |
| | M-0182907 | Germany | 1875 | *Solanum tuberosum* | 1 |
| | KM177517 | Wales | 1875 | *Solanum tuberosum* | 1 |
| | M-0182897 | USA | 1876 | *Solanum lycopersicum* | 2 |
| | M-0182906 | Germany | 1877 | *Solanum tuberosum* | 2 |
| | M-0182896 | Germany | 1877 | *Solanum tuberosum* | 2 |
| | M-0182904 | Austria | 1879 | *Solanum tuberosum* | 2 |
| | M-0182903 | Canada | 1896 | *Solanum tuberosum* | 2 |
| | KM177512 | England | NA | *Solanum tuberosum* | 1 |
| Modern samples | 06_3928A | England | 2006 | *Solanum tuberosum* | 3 |
| | DDR7602 | Germany | 1976 | *Solanum tuberosum* | 4 |
| | P1362 | Mexico | 1979 | *Solanum tuberosum* | 5 |
| | P6096 | Peru | 1984 | *Solanum tuberosum* | 5 |
| | P7722 (*P. mirabilis*) | USA | 1992 | *Solanum lycopersicum* | 5 |
| | P9464 | USA | 1996 | *Solanum tuberosum* | 5 |
| | P12204 | Scotland | 1996 | *Solanum tuberosum* | 5 |
| | P13527 | Ecuador | 2002 | *Solanum andreanum* | 5 |
| | P10127 | USA | 2002 | *Solanum lycopersicum* | 5 |
| | P13626 | Ecuador | 2003 | *Solanum tuberosum* | 5 |
| | P10650 | Mexico | 2004 | *Solanum tuberosum* | 5 |
| | LBUS5 | South Africa | 2005 | *Petunia hybrida* | 6 |
| | P11633 | Hungary | 2005 | *Solanum lycopersicum* | 5 |
| | NL07434 | Netherlands | 2007 | *Solanum tuberosum* | 3 |
| | P17777 | USA | 2009 | *Solanum lycopersicum* | 5 |
| | P17721 | USA | 2009 | *Solanum tuberosum* | 5 |

*1, Kew Royal Botanical Gardens; 2, Botanische Staatssammlung München; 3, **Cooke et al. (2012)**; 4, **Kamoun et al. (1999)**; 5, World Oomycete Genetic Resource Collection at UC Riverside, CA; 6, Dr Adele McLeod, Univ. of Stellenbosch, South Africa.

Africa (**Figure 1** and **Table 1**). These had been selected based on preliminary mtDNA information to present a cross section of *P. infestans* diversity. Finally, we sequenced two strains of *P. mirabilis*, P7722 and PIC99114, and a single strain of *P. ipomoeae*, PIC99167. Both species are closely related to *P. infestans* and served as outgroups (**Kroon et al., 2004**; **Raffaele et al., 2010a**). We considered genomes with mean-fold coverage of at least 20 as high coverage; one historic, seven modern and both outgroup genomes fulfilled this condition (**Figure 3A**). We identified single nucleotide polymorphisms (SNPs) in each sample independently by comparison with the *P. infestans* reference T30-4

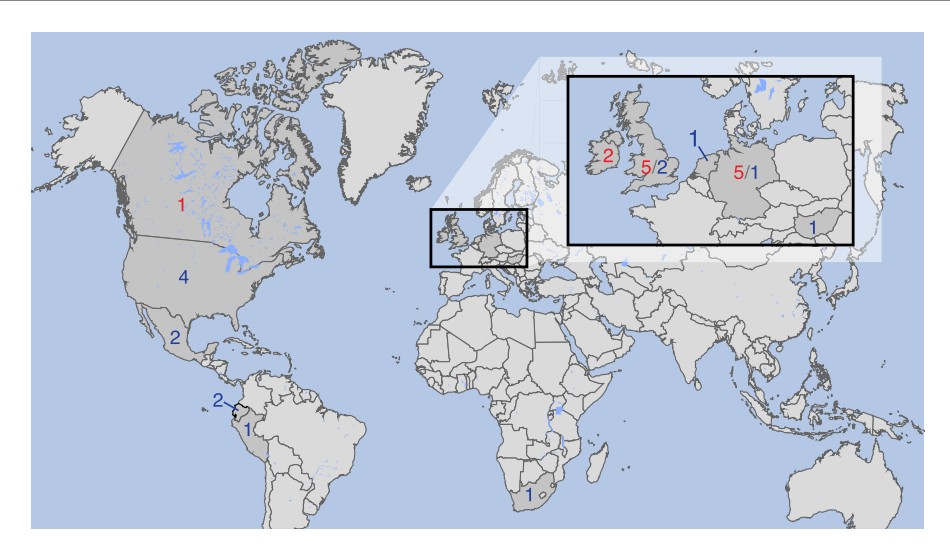

**Figure 1**. Countries of origin of samples used in whole-genome, mtDNA genome or both analyses. Red indicates number of historic and blue of modern samples. More information on the samples is given in **Tables 1 and 2**.

genome (*Figure 3B*). Thresholds for calling homozygous and heterozygous SNPs were determined from simulated data from high- and low-coverage genomes (*Figure 3—figure supplement 1*). We accepted SNPs from low-coverage genomes if the variants had also been called in a high-coverage genome. Inverse cumulative coverage plots indicated how many high- or low-coverage samples were needed to cover different fractions of SNPs (*Figure 3C,D*). A total of 4.5 million non-redundant SNPs were called. Eighty percent of all homozygous SNPs were found in at least eight samples, and only 20% of all SNPs were found in fewer than 10 strains. Thus, the great majority of polymorphic sites were shared by several strains and thus informative for phylogenetic analyses.

## A unique type I mtDNA haplotype in 19th century *P. infestans* strains

We reconstructed the mtDNA genomes from historic and modern strains using an iterative mapping assembler (*Green et al., 2008*) and estimated a phylogenetic tree from complete mtDNA genomes, with one of the *P. mirabilis* mtDNA genomes as outgroup. Previous studies have recognized four *P. infestans* mtDNA haplotype groups (Ia, Ib, IIa and IIb), based on a small number of restriction fragment length polymorphisms (RFLPs) (*Carter et al., 1990*). Surprisingly, a comparison of the complete mtDNA genomes revealed that the historic samples did not fit into any of these groups, and instead formed an independent clade, called HERB-1 from here on. Among the HERB-1 mtDNA genomes, there were very few differences, with a mean pair-wise difference of only 0.2 bp, compared to 3.9 bp for the modern haplotype I strains, and 9.0 bp for modern haplotype II strains. The origin of HERB-1 relative to haplotypes Ia and Ib could not be unequivocally resolved, and a polytomy was inferred for these three groups or support for branches were low (*Figure 4* and *Figure 4—figure supplement 1*).

The clonal lineage US-1, with the diagnostic mtDNA haplotype Ib, was the predominant genotype throughout the world until about 1980 (*Goodwin et al., 1994*). The two US-1 representatives in our material, DDR7602 (Germany) and LBUS5 (South Africa), clustered together with the Ib reference genome and samples P6096 (Peru) and P1362 (Mexico) (*Figure 4* and *Figure 4—figure supplement 1*), even though these last two samples had not been classified before as US-1 isolates. Although the US-1 genotype is closely related to the herbarium strains, US-1 is not a derivative of HERB-1. Rather, HERB-1 and US-1 are sister groups that share a common ancestor. There are three private substitutions that define the US-1 clade, and two that define the HERB-1 clade. In agreement with the previous report by *Ristaino et al. (2001)*, all historic samples had a T at the position diagnostic for haplotype Ib (*Figure 4—figure supplement 2*), which distinguishes them from the US-1 lineage, which carries instead a C at this position. In contrast to the previous report (*Ristaino et al., 2001*), we found no other sequence differences around this diagnostic site.

**Table 2.** Sequencing strategy

| ID | Instrument and read type | Sequencing center | Coverage |
|---|---|---|---|
| M-0182896 | HiSeq 2000 (2 × 101 bp) | MPI | High |
| M-0182897 | HiSeq 2000 (2 × 101 bp) | MPI | Low* |
| M-0182898 | HiSeq 2000 (2 × 101 bp) | MPI | Low |
| M-0182900 | HiSeq 2000 (2 × 101 bp) | MPI | Low† |
| M-0182903 | HiSeq 2000 (2 × 101 bp) | MPI | Low |
| M-0182904 | HiSeq 2000 (2 × 101 bp) | MPI | Low* |
| M-0182906 | HiSeq 2000 (2 × 101 bp) | MPI | Low† |
| M-0182907 | HiSeq 2000 (2 × 101 bp) | MPI | Low |
| KM177497 | MiSeq (2 × 150 bp) | MPI | Low |
| KM177500 | MiSeq (2 × 150 bp) | MPI | Low* |
| KM177502A | MiSeq (2 × 150 bp) | MPI | Low* |
| KM177507 | MiSeq (2 × 150 bp) | MPI | Low* |
| KM177509 | MiSeq (2 × 150 bp) and HiSeq 2000 (2 × 101 bp) | MPI | Low |
| KM177512 | MiSeq (2 × 150 bp) and HiSeq 2000 (2 × 101 bp) | MPI | Low |
| KM177513 | MiSeq (2 × 150 bp) and HiSeq 2000 (2 × 101 bp) | MPI | Low |
| KM177514 | MiSeq (2 × 150 bp) and HiSeq 2000 (2 × 101 bp) | MPI | Low |
| KM177517 | MiSeq (2 × 150 bp) and HiSeq 2000 (2 × 101 bp) | MPI | Low |
| KM177548 | MiSeq (2 × 150 bp) and HiSeq 2000 (2 × 101 bp) | MPI | Low |
| 06_3928A | GAIIX (2 × 76 bp) | TSL | High |
| DDR7602 | GAIIX (2 × 76 bp) | TSL | High |
| LBUS5 | GAIIX (2 × 76 bp) | TSL | High |
| NL07434 | GAIIX (2 × 76 bp) | TSL | High |
| P10127 | HiSeq 2000 (2 × 101 bp) | MPI | Low |
| P10650 | HiSeq 2000 (2 × 101 bp) | MPI | Low |
| P12204 | HiSeq 2000 (2 × 101 bp) | MPI | Low |
| P13527 | GAIIX (2 × 76 bp) | TSL | High |
| P1362 | HiSeq 2000 (2 × 101 bp) | MPI | Low |
| P13626 | GAIIX (2 × 76 bp) | TSL | High |
| P11633 | HiSeq 2000 (2 × 101 bp) | MPI | Low |
| P17721 | HiSeq 2000 (2 × 101 bp) | MPI | Low |
| P17777 | GAIIX (2 × 76 bp) | TSL | High |
| P6096 | HiSeq 2000 (2 × 101 bp) | MPI | Low |
| P7722 | HiSeq 2000 (2 × 101 bp) | MPI | Low |
| P9464 | HiSeq 2000 (2 × 101 bp) | MPI | Low* |
| PIC99114 | GAIIX (2 × 76 bp) | TSL | High |
| PIC99167 | GAIIX (2 × 76 bp) | TSL | High |

*Samples not included in any analysis due to extremely low coverage.
†Samples used only in mtDNA analysis.

## Relationship between HERB-1 and modern strains and divergence times

As the HERB-1 strains were sampled in the 19th century, their genomes should harbor fewer substitutions compared to modern strains, which have continued to evolve for over a hundred years. This can be exploited to directly calculate substitution rates and divergence times using the sampling age as tip calibration in a Bayesian framework analysis. Shorter evolutionary time usually translate into branch shortening in phylogenetic trees that include both modern and ancient pathogen strains (***Bos et al.,***

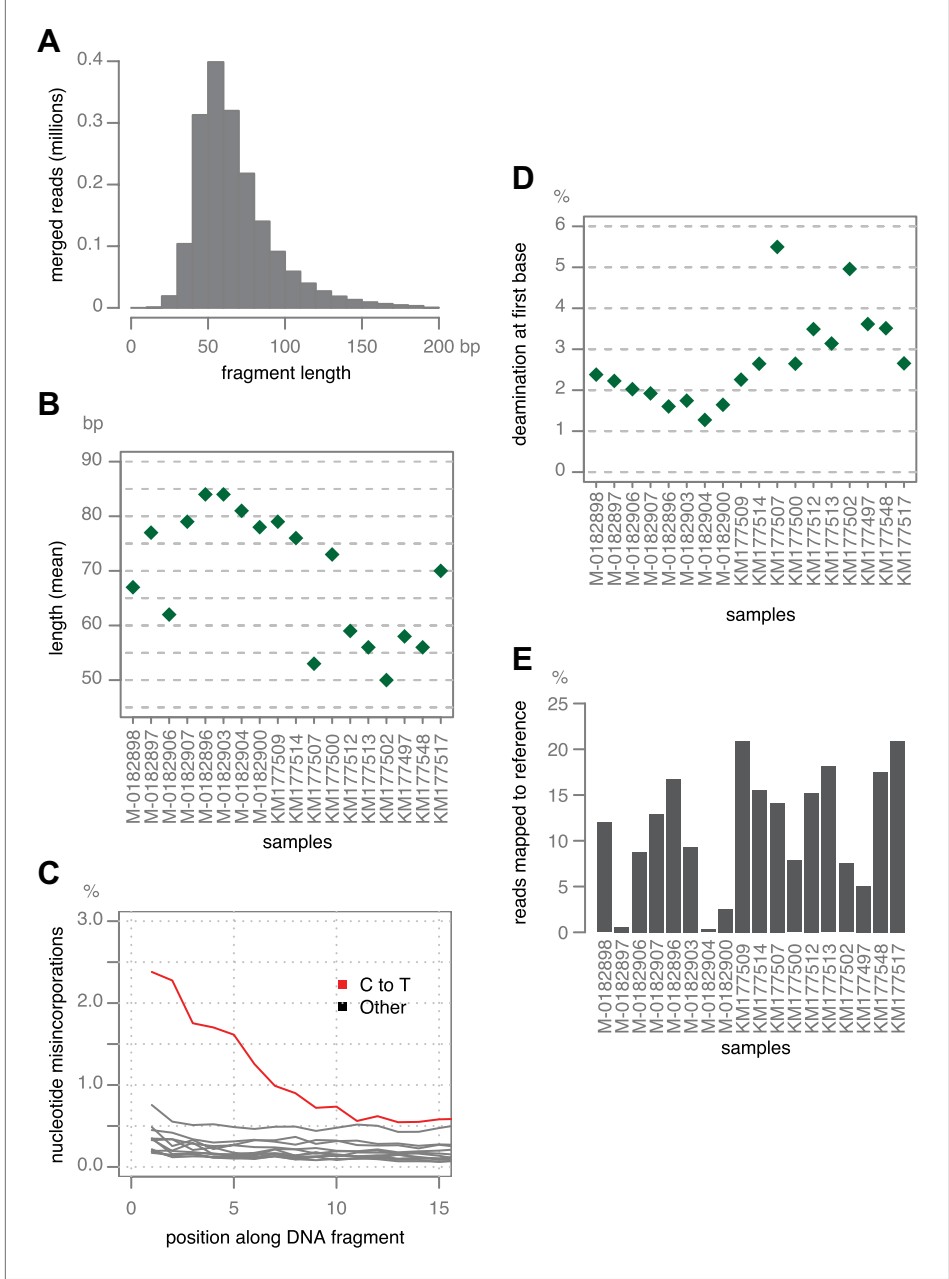

Figure 2. Ancient DNA-like characteristic of historic samples. (A) Lengths of merged reads from historic sample M-0182898. (B) Mean lengths of merged reads from historic samples. (C) Nucleotide mis-incorporation in reads from the historic sample M-0182898. (D) Deamination at first 5' end base in historic samples. (E) Percentage of merged reads that mapped to the *P. infestans* reference genome.

*2011*). By calculating the nucleotide distance to the outgroup P17777, all HERB-1 strains were found to show significantly fewer mtDNA substitutions than modern strains with haplotype Ia or Ib (p=0.0003). Sampling age of the strain and the number of mtDNA substitutions were highly correlated ($r^2$ = 0.8; *Figure 5*).

Given the correlation between sample age and the number of mtDNA substitutions, a multiple sequence alignment of 12 nearly complete modern and the 13 HERB-1 mtDNA genomes was used as input for a Bayesian framework analysis using algorithms implemented in the software package Beast (*Drummond et al., 2012*). The molecular clock hypothesis for the modern strains could not be rejected

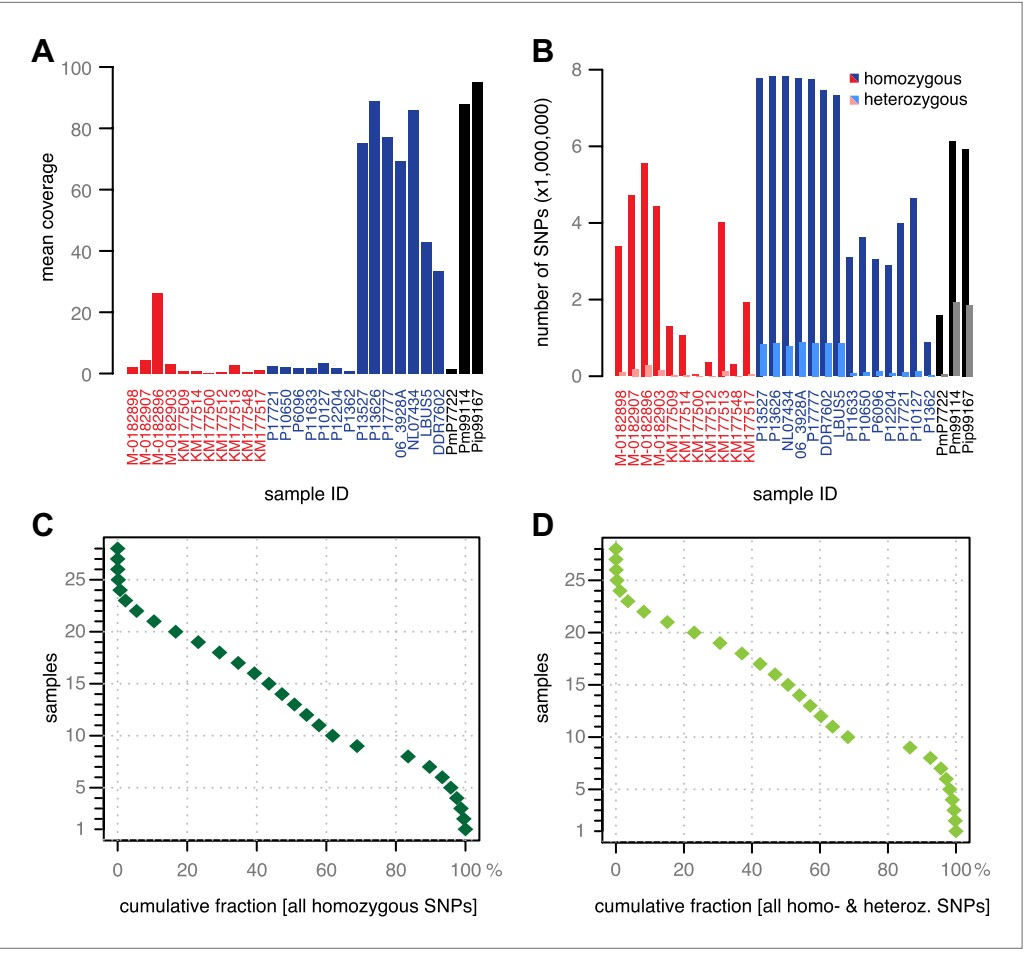

**Figure 3.** Coverage and SNP statistics. (**A**) Mean nuclear genome coverage from historic (red) and modern (blue) samples. (**B**) Homo- and heterozygous SNPs in each sample. (**C**) Inverse cumulative coverage for all homozygous SNPs across all samples. (**D**) Same as (**C**) for homo- and heterozygous SNPs.
The following figure supplements are available for figure 3:

**Figure supplement 1.** Accuracy and sensitivity of SNP calling at different cutoffs for SNP concordance based on 3- and 50-fold coverage of simulated data.

at a 5% significance level (p=0.12). Therefore, a strict molecular clock and a birth-death tree prior (*Stadler, 2010*) were used for the Bayesian framework analysis. Collection dates for all herbaria samples and the isolation dates for all modern strains were used as tip calibration points, so that substitution rates per time interval could be calculated (*Table 1*). Three Markov Chain Monte Carlo (MCMC) runs with 147 million iterations were carried out. Stability of the estimated prior and posterior probability distributions (ESS values >5000) and likelihood values (ESS values >9000) were observed in the trace files throughout the merged iterations using the software Tracer (*Rambaut and Drummond, 2007*). From this procedure, we estimated the mtDNA substitution rate to be $2.4 \times 10^{-6}$ per site and year ($1.5$–$3.3 \times 10^{-6}$, 95% HPD). This rate resulted in a mean divergence time for *P. infestans* and *P. mirabilis* of 1318 years ago (ya) (853–1836 ya 95% HPD) and for *P. infestans* type I and type II mtDNA haplotypes of 460 ya (300–643 ya 95% HPD). The origin of the 19th century herbarium clade was estimated to around 182 ya (168–201 ya 95% HPD) (*Figure 6* and *Table 3*).

To understand the evolutionary relationships between historical and modern strains in more detail, we also compared their nuclear genomes. We built phylogenetic trees with high-coverage genomes using maximum parsimony (*Figure 7A*), maximum likelihood (*Figure 7B*) and neighbor-joining (*Figure 7—figure supplement 1*) methods. We included in the analysis heterozygous biallelic

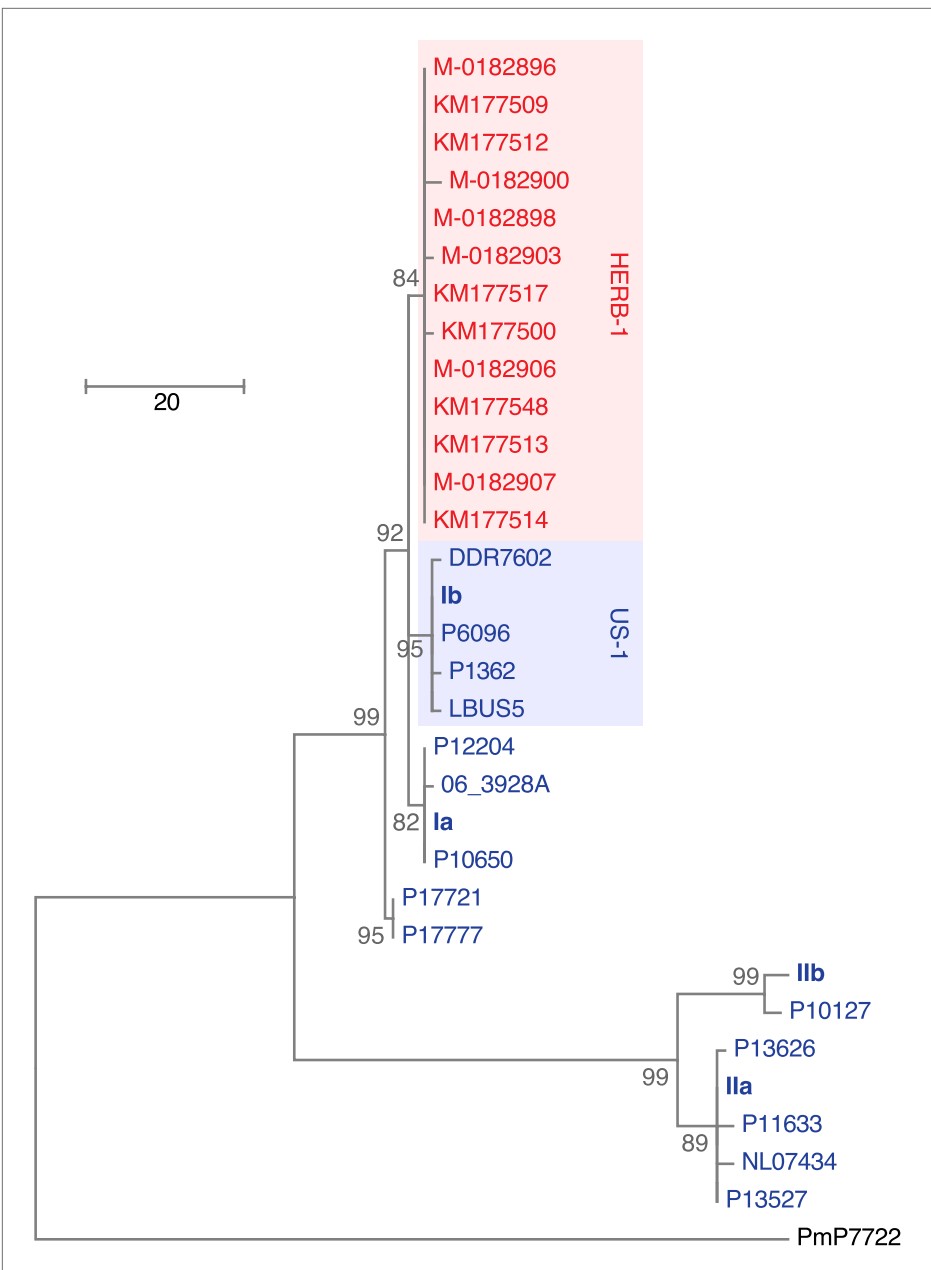

**Figure 4**. Maximum-parsimony phylogenetic tree of complete mtDNA genomes. Sites with less than 90% informa-
tion were not considered, leaving 24,560 sites in the final dataset. Numbers at branches indicate bootstrap support
(100 replicates), and scale indicates changes.

The following figure supplements are available for figure 4:

**Figure supplement 1**. Maximum-likelihood phylogenetic tree of complete mtDNA genomes.

**Figure supplement 2**. mtDNA sequences around diagnostic *Msp1* restriction site (grey) for reference haplotype
modern strains (blue) and historic strains (red).

SNPs by random sampling an allele from each of them. In all cases, the HERB-1 representative,
M-0182896, formed a distinct, isolated clade that appeared as a robust sister group to US-1 isolates
DDR7602 and LBUS5. As a caveat, the random sampling of SNPs at heterozygous positions will
inflate bootstrap support. Nevertheless, a heat map (*Figure 7C*) highlights that the two US-1 strains

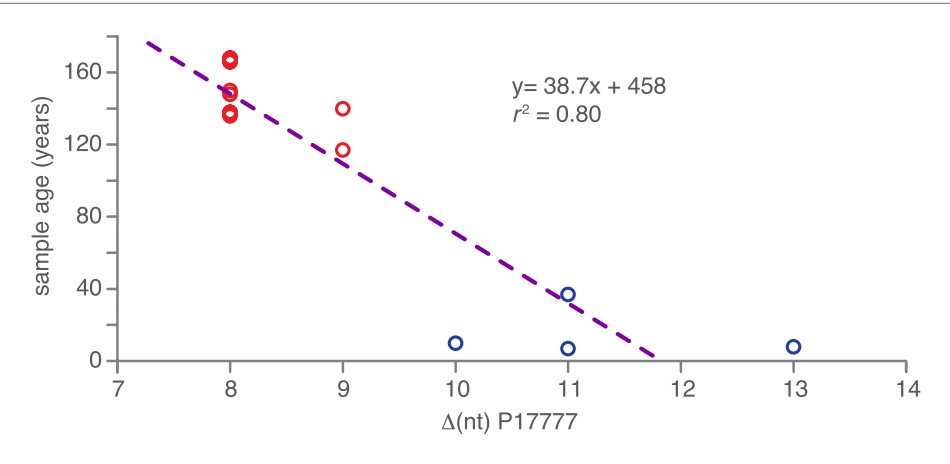

**Figure 5**. Correlation between nucleotide distance of mtDNA genomes of HERB-1/haplotype Ia/haplotype Ib clade to the outgroup P17777 and sample age in calendar years before present.

are both most closely related to HERB-1 and the most distinct among modern strains. Phylogenetic analyses that included the low-coverage genomes (*Figure 7—figure supplement 1*) were consistent with a close relationship between the HERB-1 and US-1 lineages.

## Ploidy increase in modern strains

The independent diversification of the pandemic HERB-1 and US-1 lineages together with a very recent common ancestor are consistent with both lineages having originated from the same metapopulation. To test whether the global replacement of HERB-1 by US-1 in the 20th century was due to adaptive mutations, we searched for positively selected genes using PAML (*Yang, 2007*). We did not find any evidence for genes or sites that had experienced branch-specific positive selection in any of the lineages, only a mosaic pattern with potentially positively selected genes shared across lineages. Alternative scenarios could be that inactivating mutations were more important than non-synonymous substitutions, or that higher overall diversity and re-assortment of beneficial gene variants by recombination contributed to the success of US-1.

Genetic diversity can be increased by polyploidy, which has been reported in isolates of *P. infestans* (e.g., *Daggett et al., 1995*; *Catal et al., 2010*), and which has major evolutionary implications for asexual organisms. To estimate ploidy level in our specimens, we investigated the distribution of read counts at biallelic SNPs for high-coverage genomes. In a diploid species, the mean of read counts at heterozygous positions should have a single mode at 0.5, while there should be two modes, 0.33 and 0.67, for triploid genomes, and three modes, 0.25, 0.5 and 0.75 for tetraploid genomes. We compared the observed distributions of read counts with computational simulations of diploid, triploid and tetraploid genomes. Based on the shape and kurtosis of the distributions we concluded that the historic M-0182896 genome was apparently diploid. Of the modern genomes, only NL07434 and P17777 were diploid, whereas the majority, including the two US-1 strains DDR7602 and LBUS5 as well as P13527 and P13626 were triploid. One strain, 06_3928A, even seemed to be tetraploid (*Figure 8A,B* and *Figure 8—figure supplement 1*). This conclusion was supported by polyploid strains having evidence for triallelic polymorphism at many more sites than M-0182896 (*Figure 8C*).

To further confirm the ploidy inferences, we classified 40,352 SNPs as derived or ancestral based on information from the outgroup species *P. mirabilis* and *P. ipomoeae*. We then compared the rate of homozygosity at derived alleles in M-0182896 and DDR7602. In agreement with the ploidy difference, M-0182896 had more than twice as many derived homozygous SNPs (8375) than DDR7602 (3440), regardless of annotation as synonymous, non-synonymous and non-sense (*Figure 9A,B*).

## Effector genes

*Phytophthora infestans* secretes a large repertoire of effector proteins, some of which are recognized by plant immune receptors encoded by *R* genes (*Haas et al., 2009*; *Vleeshouwers et al., 2011*).

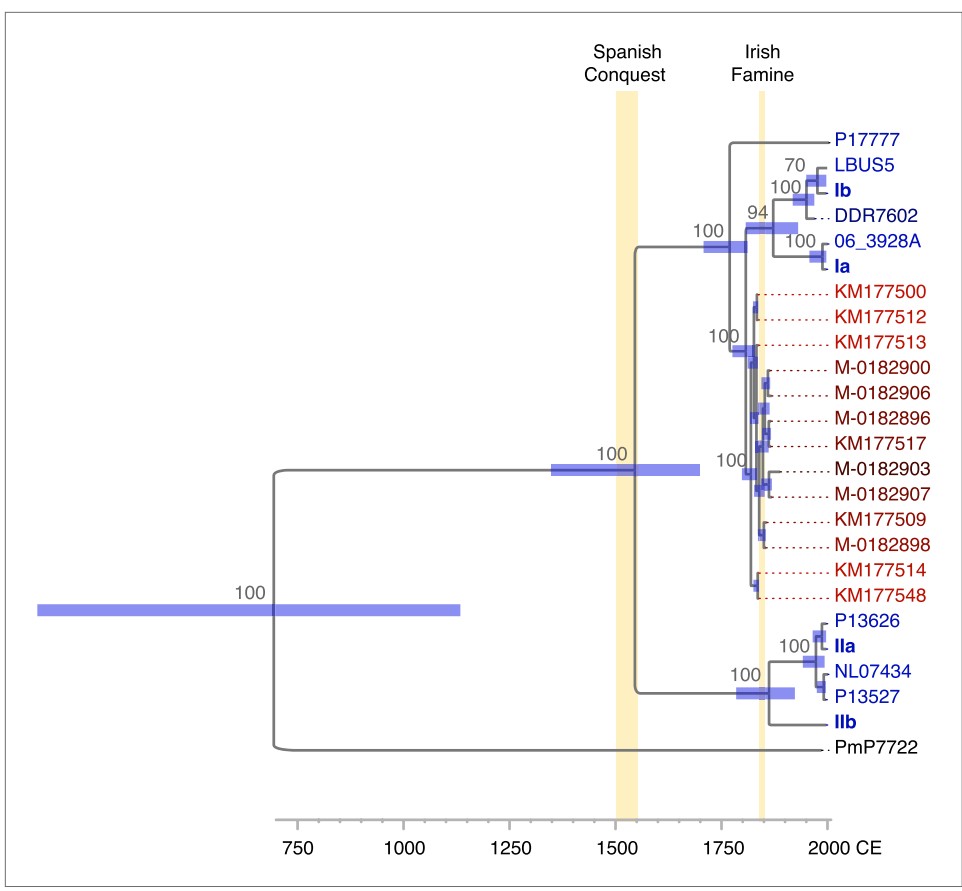

**Figure 6**. Divergence estimates of mtDNA genomes. Bayesian consensus tree from 147,000 inferred trees. Posterior probability support above 50% is shown next to each node. Blue horizontal bars represent the 95% HPD interval for the node height. Light yellow bars indicate major historical events discussed in the text. See **Figure 5** and **Table 3** for detailed estimates at the four main nodes in *P. infestans*.

These *R* genes occur in wild potato (*Solanum*) species mostly originating from the pathogen center of diversity in Mexico, and have been introduced by breeding into cultivated potato since the beginning of the 20th century (**Hawkes, 1990**). The analysis of effector gene sequences in HERB-1 strains should reveal the effector repertoire prior to its disruption by the selective forces imposed by resistance gene breeding. Given that 19th century potato cultivars in North America and Europe were fully susceptible to late blight, we presume that they did not yet contain resistance genes that are effective against HERB-1. Conversely, the first *R* genes for *P. infestans* resistance, introduced into cultivated potato only after the dates for our HERB-1 samples, should be effective against HERB-1 strains, which are predicted to carry matching avirulence effector genes. The *R* genes include in particular *R1* to *R4* from *Solanum demissum* (**Hawkes, 1990**).

To date, 10 avirulence effector genes, recognized by 10 matching *Solanum R* genes, have been described in *P. infestans* (**Vleeshouwers et al., 2011**). We first estimated the presence/absence profiles of these effector genes based on the fraction of gene length that was covered by reads ('breadth of coverage') in each high-coverage genome, and by merged reads from low-coverage genomes (**Table 4**). We deduced the amino acid

**Table 3.** Inferred time to most recent common ancestor (TMRCA) for different splits in the mtDNA tree

| Node | TMRCA (ya) | | |
| --- | --- | --- | --- |
| | Best estimate | Lower 2.5% | Upper 2.5% |
| I/HERB-1, II | 460 | 300 | 643 |
| Ia/Ib, HERB-1 | 234 | 187 | 290 |
| HERB-1 strains | 182 | 168 | 201 |
| IIa, IIb | 142 | 78 | 214 |

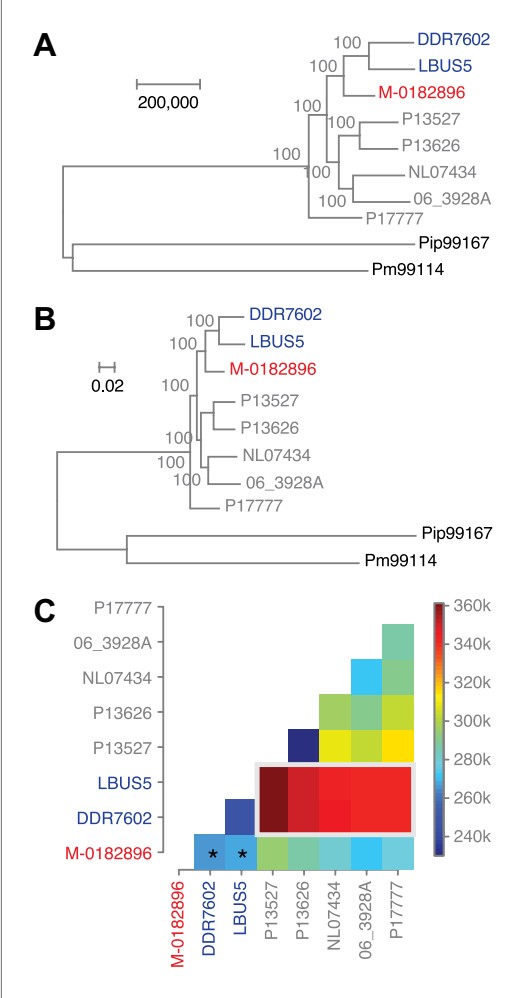

**Figure 7**. Phylogenetic trees of high-coverage nuclear genomes using both homozygous and heterozygous SNPs. (**A**) Maximum-parsimony tree, considering only sites with at least 95% information, leaving 4,498,351 sites in the final dataset. Numbers at branches indicate bootstrap support (100 replicates), and scale indicates genetic distance. (**B**) Maximum-likelihood tree. (**C**) Heat map of genetic differentiation (color scale indicates SNP differences). US-1 strains DDR7062 and LBUS5 have the genomes sequences closest to M-0182896 (asterisks). The two US-1 isolates in turn are outliers compared to all other modern strains (highlighted by a gray box).

The following figure supplements are available for figure 7:

**Figure supplement 1**. Phylogenetic trees of high- and low-coverage nuclear genomes.

sequences of these 10 effectors using both alignments of reads to the reference genome and de novo assemblies. All examined avirulence effector genes except *Avr3b* were present as full-length and intact coding sequences in the historic samples (*Table 4*), without any frame shift or nonsense mutations. The HERB-1 alleles of *Avr1*, *Avr2*, *Avr3a* and *Avr4* were shared with those of the US-1 strain DDR7602 (*Table 5—source data 1*). In conclusion, the *Avr1*, *Avr2*, *Avr3a* and *Avr4* alleles of HERB-1 are intact, presumably functional copies that are identical to ones that can be recognized by the matching *R* genes *R1*, *R2*, *R3a* and *R4* (*Armstrong et al., 2005*; *van Poppel et al., 2008*; *Gilroy et al., 2011*; *Vleeshouwers et al., 2011*). This is consistent with the expectation that the HERB-1 genotype must have been avirulent on the first potato cultivars that acquired late blight resistance.

We examined in more detail *Avr3a*, which is recognized by *Solanum demissum R3a*. The two major *Avr3a* alleles encode secreted proteins that differ in two amino acids in their effector domains: AVR3a$^{KI}$ and AVR3a$^{EM}$ (*Figure 10A*, and *Figure 10—figure supplement 1*). Only the AVR3a$^{KI}$ type triggers signaling by the resistance protein R3a (*Armstrong et al., 2005*). The *R3a* gene was introduced into modern potato from *S. demissum* at the beginning of the 20th century, providing modern potato with resistance against the *P. infestans* strains prevalent at the time (*Hawkes, 1990*; *Gebhardt and Valkonen, 2001*; *Huang et al., 2005*). Strains homozygous for *Avr3a$^{EM}$*, which avoids *R3a*-mediated detection of the pathogen, appeared later; US-1 isolates lack *Avr3a$^{EM}$* (*Armstrong et al., 2005*). Examination of *Avr3a* SNPs in the historic samples only revealed the AVR3a$^{KI}$ allele, whereas both alleles were present in modern samples (*Figure 10B*). To confirm that the potato hosts of the historic strains lacked the ability to recognize AVR3a$^{KI}$, we assembled de novo short reads from the historic samples and aligned them against the *R3a* sequence from modern potato (*Huang et al., 2005*). As expected, we only found *R3* homologs that were distinct in sequence from *R3a* (*Figure 10C*).

The absence of the *Avr3b* effector gene in HERB-1 could be viewed as puzzling, given that the *S. demissum R3* locus was one of the first to be bred into potato. However, *R3b*, the matching *R* gene of *Avr3b*, is within 0.4 cM of *R3a* in the complex *R3* locus (*Li et al., 2011*). Based on the absence of an *Avr3b* gene in HERB-1, we conclude that initial introgression of the *R3* locus from *S. demissum* was based on the resistance phenotype conferred by the *R3a* gene. The *R3* phenotype scored during the initial introgression must have been the recognition of *Avr3a* by *R3a*, and the presence of *R3b* must have been irrelevant until *P. infestans* strains carrying *Avr3b* emerged.

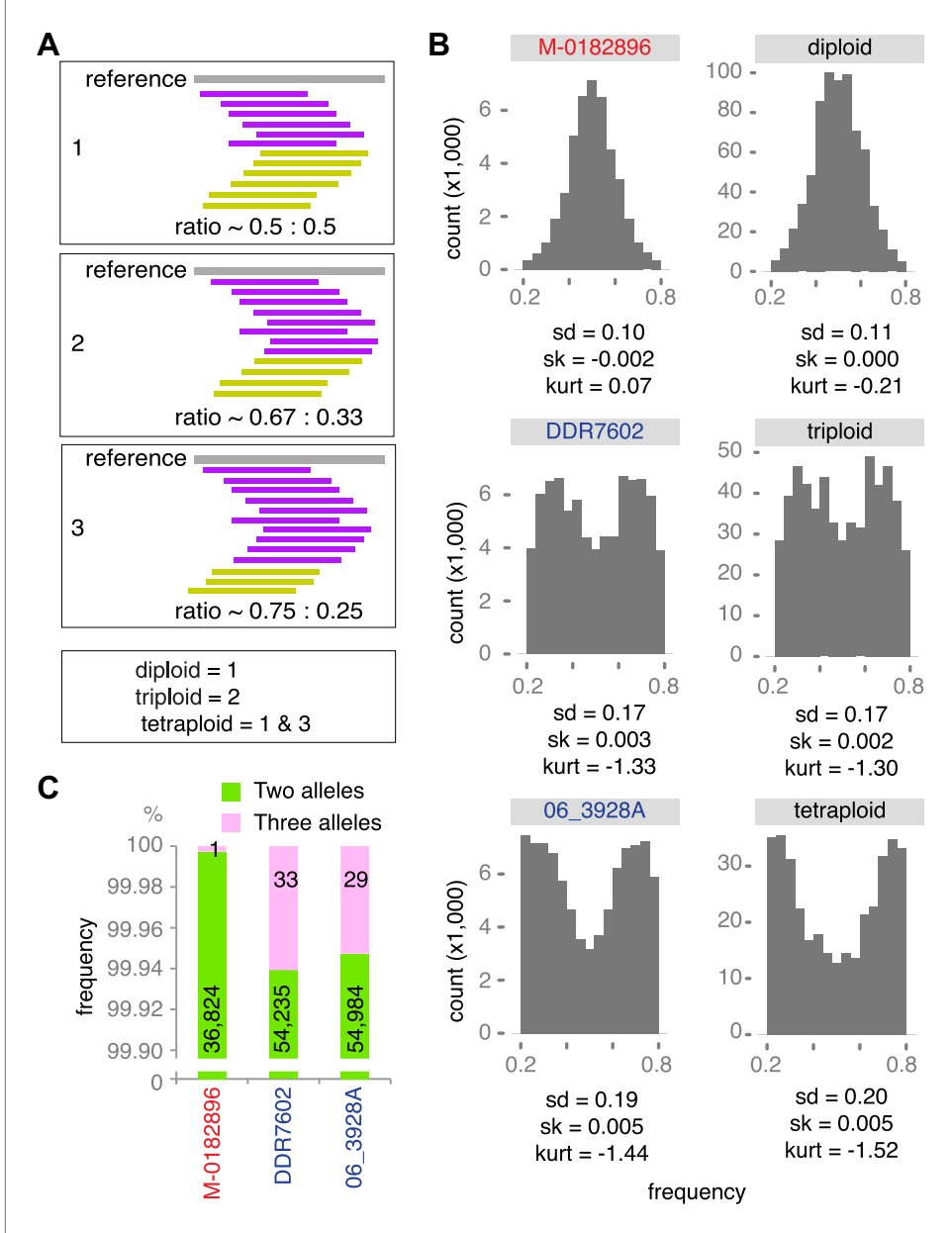

**Figure 8**. Ploidy analysis. (**A**) Diagram of expected read frequencies of reads at biallelic SNPs for diploid, triploid and tetraploid genomes. (**B**) Reference read frequency at biallelic SNPs in gene dense regions (GDRs) for the historic sample M-0182896, two modern samples, and simulated diploid, triploid and tetraploid genomes. The simulated tetraploid genome is assumed to have 20% of pattern 1 and 80% of pattern 3 shown in (**A**). The shape and kurtosis of the observed distributions are similar to the corresponding simulated ones. (**C**) Polymorphic positions with more than one allele in the GDR.

The following figure supplements are available for figure 8:

**Figure supplement 1**. Reference read frequency at biallelic SNPs in gene dense regions (GDRs) for five modern high-coverage samples.

## Discussion

To characterize the *P. infestans* strain(s) that triggered the Irish potato famine, we have sequenced several mtDNA and nuclear genomes of 19th century *P. infestans* strains. Because we wanted to interpret our findings in the context of extant genetic diversity, we investigated several modern strains as

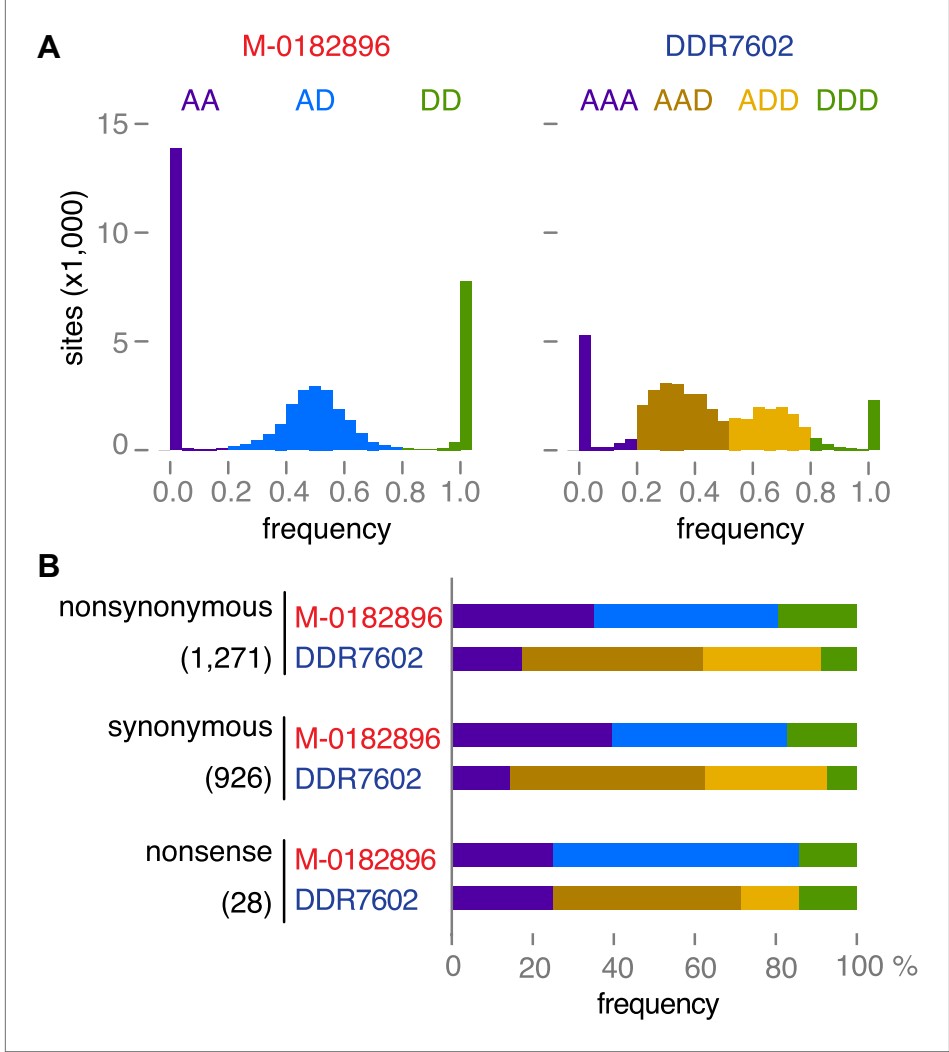

**Figure 9**. Read allele frequencies of historic genome M-0182896 and US-1 isolate DDR7602. Alleles were classified as ancestral or derived using outgroup species *P. mirabilis* and *P. ipomoeae*. There were 40,532 segregating sites. (**A**) Distributions of derived alleles at sites segregating between M-0182896 and DDR7602. (**B**) Annotation of the different site classes.

well. We could reconstruct the phylogenetic relationship between historic and modern strains using dozens of variants in complete mtDNA genomes, and millions of SNPs in the nuclear genomes, compared to previous work that had to rely on three mtDNA SNPs (*Ristaino et al., 2001*; *May and Ristaino, 2004*). The topologies of mtDNA and nuclear phylogenies were very similar, with the nuclear genomes yielding additional insights into differences in heterozygosity, ploidy levels and effector gene complement between historic and modern strains. Contrary to previous inferences (*Ristaino et al., 2001*; *May and Ristaino, 2004*), the 19th century strains are closely related to the modern US-1 lineage, but are characterized by a single, distinct mtDNA haplotype, HERB-1. Finally, from estimates of the divergence times of the different lineages, we were able to associate key events in *P. infestans* evolution with historic records of human migration and late blight spread.

## Relationship between historical and modern strains

Historic strains from different geographic locations all carried a mtDNA haplotype, HERB-1, that had not been recognized before (*Figure 4*). Although collected over a period of 50 years, the strains were distinguished from each other by few nuclear SNPs, indicating that the 19th century outbreak was a true pandemic of a rapidly spreading clonal genotype. That HERB-1 has so far not been found in any

**Table 4.** Presence or absence of avirulence effector genes in historic and modern samples, expressed as percentages of effector genes covered by reads

| | | | | 20th century non-US-1 | | | | | Outgroups | | |
|---|---|---|---|---|---|---|---|---|---|---|---|
| *Avr* gene | *R* gene | HERB-1* | US-1† | EC3527 | EC3626 | P17777 | 06_3928A | NL07434 | Merged | Pm PIC99114 | Pip PIC99167 |
| Avr1 | R1 | 100 | 100 | 0 | 0 | 100 | 0 | 0 | 100 | 98 | 100 |
| Avr2 | R2 | 100 | 100 | 100 | 100 | 100 | 81 | 100 | 77 | 97 | 100 |
| Avr3a | R3a | 100 | 100 | 100 | 100 | 100 | 100 | 100 | 100 | 0 | 28 |
| Avr3b | R3b | 0 | 0 | 0 | 0 | 100 | 0 | 0 | 100 | 100 | 100 |
| Avr4 | R4 | 100 | 100 | 100 | 100 | 95 | 89 | 100 | 99 | 85 | 92 |
| Avrblb1 | Rpi-blb1 | 100 | 100 | 100 | 100 | 100 | 100 | 100 | 100 | 0 | 0 |
| Avrblb2 | Rpi-blb2 | 100 | 100 | 100 | 100 | 92 | 100 | 100 | 89 | 88 | 0 |
| Avrvnt1 | Rpi-vnt1 | 100 | 100 | 100 | 100 | 100 | 100 | 100 | 100 | 100 | 100 |
| AvrSmira1 | Rpi-Smira1 | 100 | 100 | 100 | 100 | 100 | 100 | 100 | 100 | 97 | 100 |
| AvrSmira2 | Rpi-Smira2 | 100 | 100 | 100 | 100 | 100 | 100 | 100 | 100 | 100 | 0 |

Sequences and polymorphisms are shown in **Table 5** and **Table 5—source data 1**.
*Same sequences obtained for M-0182896 and merged sequences.
†Same sequences obtained for DDR7602 and LBUS5.

modern strain may point to its extinction after the 19th century pandemic, possibly associated with the onset of resistance gene breeding in the 20th century. We cannot, however, exclude that HERB-1 still infects some localized, genetically unimproved host populations, since we have explored only a fraction of current *P. infestans* genetic diversity. With the diagnostic variants we have discovered, one can now probe modern populations to determine whether or not HERB-1 still persists somewhere.

Historic pathogen samples are molecular fossils that can be used as tip calibration points to estimate major divergence events in the evolution of a pathogen (**Bos et al., 2011**). Using the collection dates of the herbarium samples and isolation dates of the modern *P. infestans* strains, we estimated that type I and type II mtDNA haplotypes diverged close to the beginning of the 16th century (**Figure 6** and **Table 3**). This coincides with the first contact between Americans and Europeans in Mexico, which potentially fuelled *P. infestans* population migration and expansion outside its center of diversity. This major event in human history might thus have been responsible for wider dissemination of the *P. infestans* pathogen in the New World, several centuries before its introduction to Europe. In addition, the divergence estimates allowed us to date the split between *P. mirabilis* and *P. infestans* about 1300 ya. Even though this was firmly during the period of pre-Columbian civilization, what led to their relatively recent speciation remains unknown.

To test the congruence of mtDNA and nuclear phylogenies, we reconstructed phylogenies with over 4 million nuclear SNPs from high-quality genomes (**Figure 7**). This confirmed the historic sample M-0182896 as a sister group to US-1 strains, a conclusion that was supported by a broader analysis that included the low-coverage historic samples (**Figure 7—figure supplement 1**). The private SNPs shared by the HERB-1 lineage ruled out that US-1 isolates are, as previously proposed (**Goodwin et al., 1994**), direct descendants of the historic strains. Nevertheless, US-1 is more closely related to the historic strains than to the modern strains that have come to dominate the global population in the past two decades.

We therefore propose a revision of the previous scenario, which posited that a single *P. infestans* lineage migrated around 1842 or 1843 from Mexico to North America, from where it was soon transferred to Europe, followed by global dissemination and persistence for over hundred years (**Goodwin et al., 1994**). Our data make it likely that by the late 1970s, direct descendants of HERB-1 had either become rare or extinct. On the other hand, the close relationship between HERB-1 and US-1 suggests that the US-1 lineage originated from a similar source as HERB-1, with our divergence estimates indicating that the two lineages separated only in the early 19th century. Given the much greater genetic diversity at the species' likely origin in Mexico, it seems unlikely that HERB-1 and US-1 spread independently from this region. An alternative scenario is that a small *P. infestans* metapopulation was

**Table 5.** Amino acid differences in the avirulence effectors AVR1, AVR2, AVR3a and AVR4 encoded by the T30-4 reference genome, HERB-1 and DDR7602 (US-1)

| Position | Substitution | | | Note |
| --- | --- | --- | --- | --- |
| | T30-4 | HERB1 | DDR7602 | |
| AVR1 (PITG_16,663) | | | | |
| 80 | T | T | T, S | HERB-1 polymorphisms shared with T30-4 and DDR7602. |
| 142 | I | I, T | T | |
| 154 | V | V, A | A | |
| 185 | I | I | I, V | |
| AVR2 (PITG_22,870) | | | | |
| 31 | N | K | K | HERB-1 identical to DDR7602. |
| AVR3a (PITG_14,371) | | | | |
| 19 | S | C | C | HERB-1 identical to DDR7602; both correspond to AVR3a$^{KI}$ isoform. |
| 80 | E | K | K | |
| 103 | M | I | I | |
| 139 | M | L | L | |
| AVR4 (PITG_07,387) | | | | |
| 19 | T | T, I | T | HERB-1 polymorphisms shared with T30-4 and DDR7602. |
| 139 | L | S | L, S | |
| 221 | L | V | L, V | |
| 271 | V | F | V, F | |

IDs in parentheses refer to gene models in reference genome. Full-length sequences of deduced amino acid sequences of HERB-1 AVR1, AVR2, AVR3a and AVR4 are provided in *Table 5—source data 1*.
The following source data are available for table 5:

**Source data 1**. Full-length sequences of deduced amino acid sequences of HERB-1 AVR1, AVR2, AVR3a and AVR4.

established at the periphery of its center of origin, or even outside Mexico, possibly in North America, some time before the first global *P. infestans* pandemic. The first lineage to spread from there was HERB-1, which persisted globally for at least half a century. Subsequently, the US-1 lineage expanded and spread, replacing HERB-1 (*Figure 11*).

## Genetic and genomic differences between historic and modern strains

Host *R* genes that confer resistance to historic *P. infestans* strains, such as *R3a*, were bred into cultivated potato *Solanum tuberosum* from the wild species *S. demissum* at the beginning of the 20th century, years after our youngest historic sample had been collected in 1896. In agreement with the products of these *R* genes being able to recognize HERB-1 effectors, HERB-1 strains seem to have only the *Avr3a$^{KI}$* allele, which interacts with the product of the *R* gene *R3a* to trigger a host immune response (*Armstrong et al., 2005*; *Huang et al., 2005*). Moreover, de novo assembly of potato DNA did not provide evidence for the presence of *R3a* in the herbarium hosts, consistent with the narrative of potato breeding (*Figure 10C*; *Hawkes, 1990*). While it is uncertain when HERB-1 was displaced by the US-1 lineage, the US-1 lineage also carries only the *Avr3a$^{KI}$* allele (*Armstrong et al., 2005*). The origin of the *Avr3a$^{EM}$* allele that emerged to high frequency after the breeding of *R3a* into cultivated potatoes remains unclear.

A major genomic difference between the HERB-1 and US-1 lineages is the shift in ploidy, from diploid to triploid and even tetraploid (*Figure 8* and *Figure 8—figure supplement 1*). Polyploidization could have provided an opportunity for US-1 isolates to enhance allelic diversity in the absence of frequent sexual reproduction, and could thus have contributed to their global success. Asexual reproduction leads to an increase of deleterious mutation in the population due to a lack of meiotic recombination (*Felsenstein, 1974*). Therefore, if the major selection pressure that led to the replacement of HERB-1 by US-1 was the introduction of resistance gene breeding, greater variation at effector genes

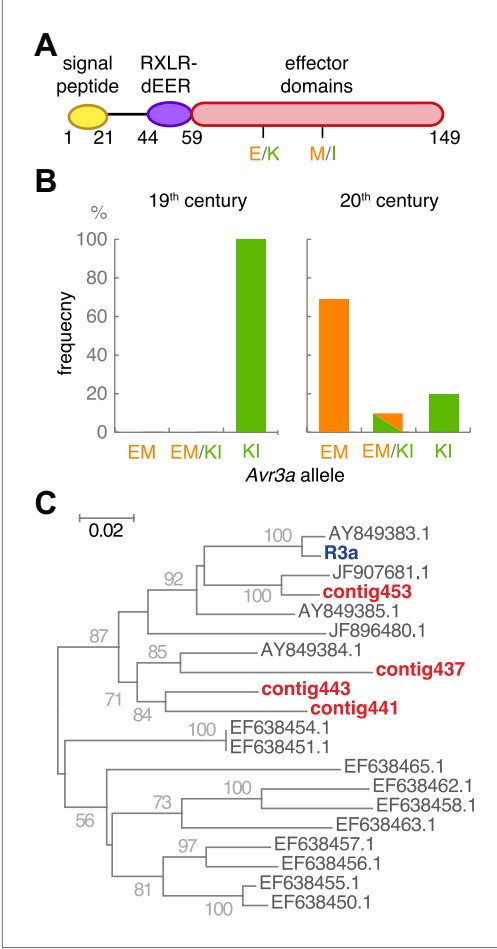

**Figure 10**. The effector gene *Avr3a* and its cognate resistance gene *R3a*. (**A**) Diagram of AVR3A effector protein. (**B**) Frequency of *Avr3a* alleles in historic and modern *P. infestans* strains. (**C**) Neighbor-joining tree of *R3a* homologs from potato, based on 0.67 kb partial nucleotide sequences of *S. tuberosum R3a* (blue, accession number AY849382.1) and homologs (dark grey) in GenBank, and de novo assembled contigs from M-0182896 (red). Numbers at branches indicate bootstrap support with 500 replicates. Scale indicates changes.
The following figure supplements are available for figure 10:

**Figure supplement 1**. Summary of de novo assembly of RXLR effector genes.

in polyploid US-1 strains could have contributed to the replacement of HERB-1 soon after *R* genes from *S. demissum* and other wild species had been introduced into modern potato germplasm.

## Conclusions

We present the first genome-wide analyses of historic plant pathogen strains. The aDNA in the herbarium samples, which were about 150 years old, was remarkably well conserved, much better than most examples of aDNA from animals and humans, and only comparable with permafrost samples (*Miller et al., 2008*; *Rasmussen et al., 2010*).

Our analyses not only highlight how knowledge of the genetics and geographic distribution of modern strains is insufficient to correctly infer the source of historic epidemics (*Goodwin et al., 1994*), but they also reveal the shortcomings of inferences that are based on a very small number of genetic markers in historic strains (*Ristaino et al., 2001*; *May and Ristaino, 2004*). With our much richer dataset, we could demonstrate that the late blight outbreaks during the 19th century were a pandemic caused by a single *P. infestans* lineage, but that this lineage was not the direct ancestor of the one that had come to dominate the global *P. infestans* population during much of the 20th century. Infected plant specimens stored in herbaria around the world are thus a largely untapped source to learn about events that affected millions of people during our recent history.

# Material and methods

## Herbarium sampling

Plant specimens were sent to the Senckenberg Museum in Frankfurt am Main by the Botanische Staatssammlung München and the Kew Royal Botanical Gardens, where potato and tomato leaves with lesions indicative of *P. infestans* infection were sampled, retrieving both the lesions and healthy surrounding tissue. Sampling was carried out under sterile conditions in a laboratory with no prior exposure to *P. infestans*. Samples were subsequently sent to the Palaeogenetics laboratory at the University of Tübingen.

## DNA extraction and sequencing library preparation

Preamplification steps of historic samples were performed in clean room facilities with no prior exposure to *P. infestans* DNA. Samples were extracted following the protocol of (*Kistler, 2012*), using 380–500 μg of each sample. Tissue was crushed with mortal and pestle, 1.2 ml extraction buffer (1% SDS, 10 mM Tris pH 8.0, 5 mM NaCl, 50 mM DTT, 0.4 mg/ml proteinase K, 10 mM EDTA, 2.5 mM *N*-phenacylthiazolium bromide) was added, and samples were incubated over night at 37°C with constant agitation. A modified protocol with the Qiagen Plant DNEasy Mini kit (Qiagen, Hilden, Germany) was then used to purify the extracted DNA.

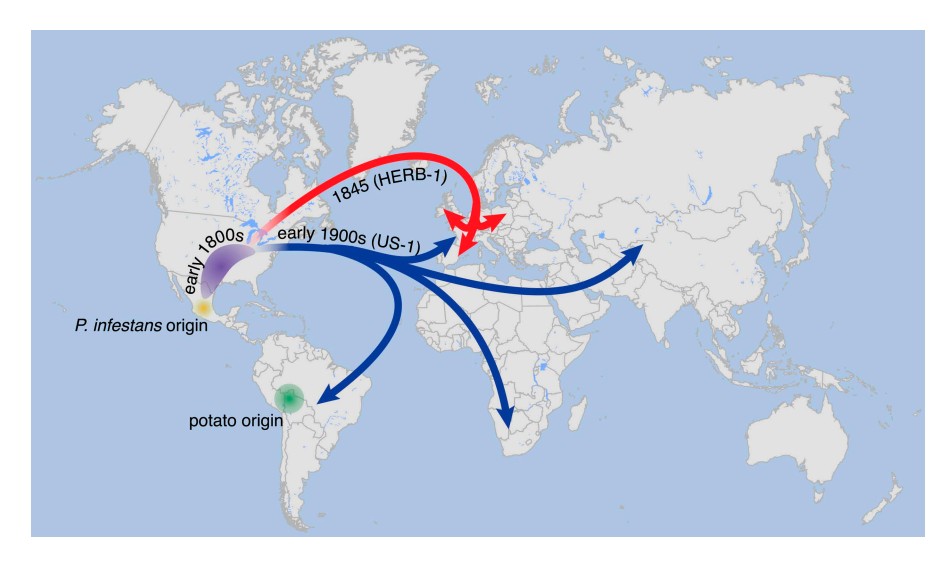

**Figure 11**. Suggested paths of migration and diversification of *P. infestans* lineages HERB-1 and US-1. The location of the metapopulation that gave rise to HERB-1 and US-1 remains uncertain; here it is proposed to have been in North America.

Two independent Illumina sequencing libraries were created for each DNA extract. In the first library, C-to-T damage caused by deamination of cytosines (*Hofreiter et al., 2001*) was not repaired. 20 µl of each DNA extract, extraction blank control and water library blank control were converted into sequencing libraries as described (*Meyer and Kircher, 2010*) with modifications for aDNA (*Meyer et al., 2012*). To avoid potential sequencing artifacts caused by DNA damage, a second library was made from, 30 µl of each DNA extract, extraction blank control and water library blank control, and treated with uracil-DNA glycosylase (UDG) and Endonuclease VIII before conversion into sequencing libraries (*Briggs et al., 2010*). Each library received sample-specific double indices after preparation via amplification with two 'index' PCR primers (*Meyer et al., 2012*). Indexed libraries were individually amplified in 100 µl reactions containing 5 µl library template, 2 units of AccuPrime Pfx DNA polymerase (Invitrogen, Karlsruhe, Germany), 1 unit of 10× PCR Mix and 0.3 µM primers spanning the index sequences of the libraries. The following thermal profile was used: 2-min initial denaturation at 95°C, two or three cycles consisting of 15 s denaturation at 95°C, a 30-s annealing at 60°C and a 2-min elongation at 68°C, and a 5-min final elongation at 68°C. Amplified products were purified and quantified on an Agilent 2100 Bioanalyzer DNA 1000 chip.

DNA extracts of the modern *P. infestans* samples P17721, P10650, P6096, P11633, P10127, P9464, P12204 and P13626 and *P. mirabilis* P7722 were obtained from the World Phytophthora and Oomycete Genetic Resource Collection, fragmented to 300 bp using a S220 Covaris instrument according to the manufacturers' protocol (Duty cycle 10%, intensity 4, cycles per burst 200, time [in s] 120), and converted into sequencing libraries following the above steps as described for the historic samples without UDG treatment (*Kircher, 2012*; *Meyer et al., 2012*). For *P. mirabilis* PIC99114 and *P. ipomoeae* PIC99167, genomic DNA used before (*Raffaele et al., 2010b*; *Cooke et al., 2012*) was converted into Illumina sequencing libraries. Genomic DNA from the other modern strains was isolated as described (*Cooke et al., 2012*).

Libraries were sequenced on Illumina GAIIx, HiSeq 2000 or MiSeq instruments (*Table 2*). To estimate the fraction of *P. infestans* aDNA in the herbarium samples, we performed pilot sequencing. Once the samples with the highest amount of *P. infestans* were identified, production runs were carried out on an Illumina HiSeq 2000 instrument. For *P. infestans* 06_3928A analysis, we used publicly available short reads (ENA ERP002420).

## Read mapping and SNP calling

Reads for the herbarium samples were de-indexed as described (*Kircher, 2012*). Forward and reverse reads were merged after adapter trimming, requiring at least 11 nucleotides overlap (*Burbano et al.,*

*2010*). Only fragments that that allowed merging of reads were used in subsequent analyses. Merged reads were mapped to the *P. infestans* T30-4 reference genome (*Haas et al., 2009*) with BWA, allowing two gaps and without seeding (*Li and Durbin, 2009*). PCR duplicates were identified based on read start and end alignment coordinates. For each cluster of duplicates a consensus sequence was calculated as described (*Kircher, 2012*). Only reads with a Phred-like mapping quality score of at least 30 were used further. Alignments were converted to BAM files (*Li et al., 2009*). For modern strains, single reads were mapped in a similar manner, and subsequent processing was performed as described (*Cooke et al., 2012*).

SNPs in the herbarium samples were called by independently comparing each strain with the *P. infestans* T30-4 genome. Raw allele counts for each position were obtained using pileup from SAMtools (*Li et al., 2009*). To decide the cutoffs for SNP calling we resorted to simulations. Reads from two 50-fold and 3-fold coverage genomes were simulated using the pIRS software (*Hu et al., 2012*), with empirical base-calling and GC%-depth profiles trained on five modern *P. infestans* genomes (P13527, P13626, 06_3928A, NL07434 and P17777). Based on the simulated data we called both homo and heterozygous SNPs employing different cutoffs for SNP concordance (*Figure 3—figure supplement 1*). Genotypes calls were classified as high quality if coverage was at least 10. We also considered low-quality SNPs, if a high-quality SNP call had been made in a different strain, using specific coverage cutoffs for rescuing low-quality SNPs (*Figure 3—figure supplement 1*). We calculated sensitivity and accuracy of SNP calls for different combination of cutoffs and selected the following criteria:

- Minimum coverage of 10 for high quality calls.
- Concordance ≥80% for homozygous SNPs.
- Concordance between 20% and 80% for heterozygous SNPs.
- Minimum coverage of 3 to rescue low-quality SNPs.

We called synonymous, nonsynonymous and nonsense polymorphisms employing snpEff (*Cingolani et al., 2012*).

## Mitochondrial DNA (mtDNA) assembly and phylogenetic analyses

Fragments that could be aligned to any of the four reference haplotypes (Ia, IIa, Ib and IIb) were used to assemble mtDNA genomes. For each strain four different assemblies (one for each reference haplotype) were built using an iterative mapping assembly program (*Green et al., 2008*; *Burbano et al., 2010*). Only positions with coverage of at least 3 were included in the assemblies. The four assemblies were aligned using Kalign (*Lassmann and Sonnhammer, 2005*) with default parameters, and a consensus assembly was generated based on the alignment.

The 1.8-kb insertion present in haplotype II was not considered for phylogenetic reconstruction. The mtDNA phylogeny was built using a maximum parsimony and a maximum likelihood tree using MEGA5 (*Tamura et al., 2011*). For both, positions with less than 90% site coverage were eliminated. There were a total of 24,560 positions in the final dataset, compared to the multiple sequence alignment length (37,762 bp). For the maximum likelihood reconstruction we used the Hasegawa-Kishino-Yano (HKY) model assuming that a certain fraction of sites are evolutionarily invariable. The model was selected using MEGA5 (*Tamura et al., 2011*).

## Nuclear genome phylogenetic analyses

We reconstructed the nuclear phylogeny for the high-coverage samples alone and for all samples together independently, using maximum parsimony and maximum likelihood approaches. We built the neighbor-joining trees based on a genetic distance matrix calculated from both homo- and heterozygous SNPs (*Xu et al., 2012*). For the high-coverage genomes we used only the SNPs positions with complete information in all samples. For the low-coverage genomes we used homo- and heterozygous SNPs, and allowing for missing data. So that we could include heterozygous SNPs in the analysis, we randomly sampled one allele at each site. The maximum parsimony trees were calculated with 100 replicates in MEGA5 using the Close-Neighbor-Interchange algorithm with search level 0, in which the initial trees were obtained with the random addition of sequences (10 replicates). All positions with less than 95% site coverage were eliminated (*Tamura et al., 2011*). For the high-coverage genomes-only analysis, all positions with less than 85% site coverage were eliminated. For the all-sample analysis the threshold was lowered to 80%. Maximum likelihood trees were built using RaxML 7.0.4 with 100 replicates using the rapid bootstrap algorithm (*Stamatakis, 2006*).

## Effector analyses

To address presence and absence polymorphisms of effectors, we used a previously published pipeline (*Raffaele et al., 2010a*). We calculated the fraction of each gene that was covered by reads ('breadth of coverage') for each strain. We regarded zero breadth of coverage as absence of the gene. For herbarium and modern samples with genome-wide coverage depth less than 20×, we merged BAM files from each strain into a single BAM file, and used this to estimated breadth of coverage.

We also tested for presence/absence polymorphisms of RXLR effector genes between herbarium samples and modern strains using de novo assembly of short reads. First, we extracted 140 bp fragments from M-0182896 merged reads, and assembled these with SOAPdenovo v1.05 (*Luo et al., 2012*). We aligned the six-frame translation of resulting contigs to a non-redundant protein database using blastx (*Altschul et al., 1990*). Most contigs matched proteins from either potato, *Solanum tuberosum*, or to microbial species *P. infestans*, *Pantoea vagans* and *Pseudomonas* sp. To focus on *P. infestans*, we aligned fragments that were at least 140 bp to the genomes of *P. infestans*, *S. tuberosum*, *P. vagans*, *P. syringae* pv. *syringae*, and *P. fluorescens* with blastn. We extracted fragments that aligned the *P. infestans* genomic regions encoding RXLR effector genes, but over at most 90 bp. These and unmapped fragments were de novo assembled with SOAPdenovo v1.05. A *k*-mer size of 67 was deemed optimal, because it resulted in the highest coverage of *Avr1*, *Avr2* and *Avr3a*, and resulted in the largest number of RXLR proteins with TBLASTN hits (*Figure 10—figure supplement 1*). We obtained partial sequences of *Avr4* and *Avrblb1*. We visually evaluated BWA alignments of M-0182896 in the *Avr4* and *Avrblb1* genomic regions and identified T30-4 sequences uncovered by alignments using Integrative Genomics Viewer (*Robinson et al., 2011*). We then identified T30-4 genomic regions with at least 99% similarity to these uncovered regions. In BWA, if reads match several genomic regions, one genomic location is randomly chosen as default (*Li and Durbin, 2009*). Thus, it is possible that BWA alignment distributes reads coming from the same gene across several, closely related genes in the target genome. We assembled such reads that mapped to closely related sequences in the reference genome together with the partial sequences of *Avr4* and *Avrblb1* using Geneious Pro 5.6.3 to obtain full-length sequences of these *Avr* genes.

## Selection tests

Homozygous SNPs from modern *P. infestans* strains EC3527, EC3626, NL07434, 06_3928A, DDR7602, LBUS5, P17777 and the historic strain M-0182896 were used for selection tests. Gene sequences were converted into amino acid sequences using EMBOSS tools (*Rice et al., 2000*), and Pla2Nal v14 (*Suyama et al., 2006*) was used to convert protein alignments to codon alignments. The codeml module of PAML package v4.6 (*Yang, 2007*) was used for positive selection studies with site models M7 (parameters NSsites = 7, fix_omega = 0, omega = 2 and kappa = 3) and M8 (NSsites = 8, fix_omega = 0, omega = 2 and kappa = 3). A 5% level of significance was established with Likelihood ratio test. Genes were considered to be under positive selection if at least one site was found to be under selection with a Bayes Empirical Bayes confidence >95%.

## Ploidy analyses

To estimate ploidy levels, we assessed the distributions of read counts at biallelic SNPs. For diploid species, the mean frequency of reads for each allele at non-homozygous sites is 1/2, while we expect two modes for triploid genomes, at 1/3 and 2/3, and four modes for tetraploid genomes, at 1/4, 1/2 and 3/4 (*Figure 8A*). We simulated genomes with different ploidy levels using pIRS (*Hu et al., 2012*), based on two strains, *P. infestans* T30-4 and EC3527. The SNPs used for the construction of two simulated chromosomes were determined with SAMtools v0.1.8 mpileup and bcftools v0.1.17 (*Li et al., 2009*). For the diploid genome, we simulated 10x coverage reads for each of two different chromosomes. For the triploid genome, we merged simulated 5× and 15× coverage reads from two different chromosomes. For the tetraploid genome, we merged simulated 10× coverage reads from two different chromosomes (*Figure 8B*). Next, we aligned the simulated reads to the *P. infestans* T30-4 reference genome with BWA and called heterozygous SNPs under the following criteria: minimum coverage of 10 for high-quality calls, and concordance between 20% and 80% for heterozygous SNPs. Since tetraploid species are considered to be a mixtures of two ratio, we mixed SNPs from the 20× coverage diploid reads and the 20× coverage tetraploid reads in following ratios: 0:100, 10:90, 20:80, 30:70, 40:60, 50:50, 60:40, 70:30, 80:20, 90:10 and 100:0. Finally, we estimated frequency of reads assigning each allele at each SNP position. Based on shapes, standard deviation, skewedness and kurtosis of the

observed distributions and comparison with the simulated distributions, we classified the tested *P. infestans* genomes as diploid, triploid and tetraploid.

## Substitution rates and divergence times for *P. infestans*

In order to test whether we can detect a temporal signal in the ancient *P. infestans* mtDNA sequences compared to modern strains, that is, shorter branches in the ancient strains compared to the modern ones, we calculated the nucleotide distance as the number of substitutions between HERB-1, haplotype Ia and haplotype Ib mtDNA genomes to the outgroup P17777. The analysis involved 19 nucleotide sequences. All positions with less than 90% site coverage were eliminated, resulting in 34,174 informative positions. The samples were subsequently grouped into ancient and modern strains. The ancient and modern nucleotide distances were significantly different (Mann–Whitney U-test, p=0.0003). We furthermore correlated the nucleotide distance of HERB-1, haplotype Ia and haplotype Ib mtDNA genomes to the outgroup P17777 with the tip age of each sample.

To estimate divergence times of *P. infestans* strains, substitution rates were calculated in a Bayesian framework analysis using the software package BEAST 1.7.5 (*Drummond et al., 2012*). A multiple sequence alignment that included all 12 nearly complete modern *P. infestans* mtDNA sequences plus all 13 herbaria samples was used as input. In order to test if the mtDNAs evolved clock like a likelihood ratio test was performed in MEGA5 (*Tamura et al., 2011*) by comparing the maximum likelihood (ML) value for the given topology using only the modern strains with and without the molecular clock constraints. The null hypothesis of equal evolutionary rate throughout the tree was not rejected at a 5% significance level (p=0.115). All positions containing gaps and missing data were eliminated resulting in a total of 22,591 positions in the final dataset.

As a result a strict molecular clock and the HKY sequence evolution model were used for the Bayesian framework. For the tree prior, five different models were tested including four coalescence models: constant size, expansion growth, exponential growth, logistic growth and a epidemiology birth–death model (*Stadler, 2010*). For each tree prior, three MCMC runs were carried out with 10,000,000 iterations each and subsequently merged using LogCombiner 1.7.5 from the BEAST package. Resulting ESS values and overall posterior likelihoods were compared using the software Tracer (*Rambaut and Drummond, 2007*). The birth-death model gave the highest ESS values and posterior likelihood and was therefore chosen for the subsequent dating analysis. The collection dates for all herbaria samples as well as the isolation dates for all modern strains were used as tip calibration points (*Table 1*). Three MCMC runs were carried out with 50,000,000 iterations each, sampling every 10,000 steps. The first 1,000,000 iterations were discarded as burn-in resulting in a total of 147,000,000 iterations.

## Acknowledgements

We are indebted to Bryn Dentinger, curator of the herbarium of the Kew Royal Botanical Gardens, and to Dagmar Triebel, curator of the herbarium of the Botanische Staatssammlung München, for providing the historic specimens used in this study. We are grateful to Mike Coffey for discussion of strain selection for genome analyses and providing genomic DNA preparations, Tahir Ali for help with phylogenetic tests, Jodie Pike for sequencing support, and Mike Coffey, David Cooke, Geert Kessel, Adele McLeod, Ricardo Oliva and Vivianne Vlesshouwers for *Phytophthora* strains. We thank Axel Künstner, Dan Koenig, Jorge Quintana and Ignacio Rubio-Somoza for discussion on data analysis and Eunyoung Chae and Rebecca Schwab for comments on the manuscript.

## Additional information

### Competing interests

DW: Deputy editor, *eLife*. The other authors declare that no competing interests exist.

### Funding

| Funder | Author |
| --- | --- |
| European Research Council | Kentaro Yoshida, Verena J Schuenemann, Liliana M Cano, Marina Pais, Sophien Kamoun, Johannes Krause |
| Gatsby Charitable Foundation | Kentaro Yoshida, Liliana M Cano, Marina Pais, Sophien Kamoun |

| Funder | Author |
|---|---|
| Biotechnology and Biological Sciences Research Council (BBSRC) | Kentaro Yoshida, Liliana M Cano, Marina Pais, Sophien Kamoun |
| LOEWE | Bagdevi Mishra, Rahul Sharma, Marco Thines |
| United States Department of Agriculture | Frank N Martin |
| Japan Society for the Promotion of Science | Kentaro Yoshida |
| Deutsche Forschungsgemeinschaft | Detlef Weigel, Hernán A Burbano |
| Max Planck Society | Chirsta Lanz, Detlef Weigel, Hernán A Burbano |

The funders had no role in study design, data collection and interpretation, or the decision to submit the work for publication.

## Author contributions

KY, Processed sequencing reads of modern strains, identified SNPs, performed phylogenetic, genome-wide and effector gene analyses, discussed interpretation and wrote the manuscript; VJS, Extracted DNA from herbarium samples and prepared genomic libraries; LMC, Processed sequencing reads of modern strains and performed genome-wide and effector genes analysis; MP, Coordinated sequencing of modern strains in Norwich; BM, RS, Performed positive selection analyses; CL, Coordinated sequencing of historic and modern strains in Tübingen; FNM, Performed mtDNA haplotype analyses for choosing modern strains for genome sequencing; SK, DW, Conceived the project, coordinated the collaborative effort, discussed interpretation and wrote the manuscript; JK, Conceived the project, coordinated aDNA wet-lab experiments, performed phylogenetic analysis and discussed interpretation and wrote the manuscript; MT, Conceived the project, coordinated the collection and sampling of herbarium strains, discussed interpretation and wrote the manuscript; HAB, Conceived the project, coordinated the collaborative effort, processed sequencing reads of modern and historic strains, identified SNPs, performed phylogenetic analysis, discussed interpretation and wrote the manuscript

# Additional files

## Major datasets

The following datasets were generated:

| Author(s) | Year | Dataset title | Dataset ID and/or URL | Database, license, and accessibility information |
|---|---|---|---|---|
| Yoshida K, Schuenemann VJ, Cano LM, Pais M, Mishra B, Sharma R, Lanz C, Martin FN, Kamoun S, Krause J, Thines M, Weigel D, Burbano HA | 2013 | Sequencing *Phytophthora infestans* genomes | ERP002419; http://www.ebi.ac.uk/ena/data/view/ERP002419 | Publicly available at the Sequence Read Archive (http://www.ncbi.nlm.nih.gov/sra). |
| Yoshida K, Schuenemann VJ, Cano LM, Pais M, Mishra B, Sharma R, Lanz C, Martin FN, Kamoun S, Krause J, Thines M, Weigel D, Burbano HA | 2013 | Resequencing *Solanaceae* (Potato and Tomato) 19th century samples | ERP002550; http://www.ebi.ac.uk/ena/data/view/ERP002550 | Publicly available at the Sequence Read Archive (http://www.ncbi.nlm.nih.gov/sra). |
| Yoshida K, Schuenemann VJ, Cano LM, Pais M, Mishra B, Sharma R, Lanz C, Martin FN, Kamoun S, Krause J, Thines M, Weigel D, Burbano HA | 2013 | Resequencing *Phytophthora* strains | ERP002552; http://www.ebi.ac.uk/ena/data/view/ERP002552 | Publicly available at the Sequence Read Archive (http://www.ncbi.nlm.nih.gov/sra). |

The following previously published datasets were used:

| Author(s) | Year | Dataset title | Dataset ID and/or URL | Database, license, and accessibility information |
|---|---|---|---|---|
| Haas BJ, Kamoun S, Zody MC, Jiang RH, Handsaker RE, Cano LM, Grabherr M, Kodira CD, Raffaele S, Torto-Alalibo T, et al. | 2009 | *Phytophthora infestans* Database, Broad Institute | http://www.broadinstitute.org/annotation/genome/phytophthora_infestans/MultiHome.html | Publicly available at http://www.broadinstitute.org/annotation/genome/phytophthora_infestans/MultiDownloads.html. |
| Cooke DEL, Cano LM, Rafffaele S, et al. | 2012 | Genome analyses of an aggressive and invasive lineage of the Irish potato famine pathogen | ERP002420; http://www.ebi.ac.uk/ena/data/view/ERP002420 | Publicly available at the Sequence Read Archive (http://www.ncbi.nlm.nih.gov/sra). |

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
