## [Decision Letter]

Thank you for sending your work entitled “Herbarium metagenomics reveals the rise and fall of the *Phytophthora* lineage that triggered the Irish potato famine” for consideration at *eLife*. Your article has been favorably evaluated by a Senior editor and 4 reviewers, one of whom is a member of our Board of Reviewing Editors.

The following individuals responsible for the peer review of your submission want to reveal their identity: David Baulcombe (Reviewing editor); Paul Birch (peer reviewer); William Fry (peer reviewer).

The reviewers agreed that this is an interesting and important paper. It draws on both the availability of historic samples of *P. infestans* and the power of next-generation sequencing to re-evaluate the 19th century pandemic that precipitated the Irish Famine. Overall, the conclusion of a single clonal lineage, HERB-1, dominating the late blight population outside of Mexico for at least 50 years, closely related to US-1, but perhaps giving way to variants derived from the latter, is well supported by the data. The derivation of both HERB-1 and US-1 from a common ancestor within a “metapopulation” outside of Mexico, the accepted centre of *P. infestans* diversity, again seems likely. Of interest is the apparent increase in ploidy levels in 20th century isolates, compared to HERB-1 and US-1, although it should be stressed that this is based entirely on projected allele frequencies (rather than backed up by other means).

In addition, there is part of the analysis that they consider to lack detail and that the conclusions may be reaching too far. These concerns are summarised by one of the reviewers who writes:

1) The authors state: “Given that nineteenth century potato strains [surely they mean cultivars?] in North America and Europe did not yet contain resistance genes to control HERB-1, one would expect HERB-1 to contain a full effector gene complement”, as it has not yet been disrupted “…by the selective forces imposed by resistance gene breeding”. What exactly is meant by a “full effector gene complement”? I guess they intend the focus to be on avirulences recognised by the *S. demissum R* genes? These would be AVR1, 2, 3a, 3b, and 4 of those shown in Table 2; the others are not pertinent to their line of logic. In Table 2, the coverage of AVR2 is 100% for the isolate 06_3928A, a “Blue_13” genotype representative. Actually, this genotype lacks AVR2, expressing instead AVR2-like, which suggests that the % coverage they show in this table is lacking the detail to reach their conclusion: “consistent with the expectation that the HERB-1 genotype was avirulent on the first potato cultivars that acquired late blight resistance through breeding”. The focus is on AVR3a, and it is interesting that the KI allele alone exists in the HERB-1 lineage. Did they specifically amplify and sequence this gene from all of the herbarium samples? If their conclusions are based entirely on sequence assemblies (from NGS of ancient DNA), then they need to be backed up by additional experiments. AVR4 has been well documented to be “lost” through mutation that leads to truncated proteins. Is AVR4 intact in the historical samples? I am surprised by the apparent absence of *Avr3b*. This is a gene that is conserved in *P. parasitica*, for example. Indeed, they show good coverage in mirabilis and ipomeae in Table 2. Did they specifically check this absence by alternative means (i.e., PCR and sequencing)?

2) The reviewers believe that these concerns can be addressed in a straightforward fashion, if you amplify and directly study AVR3a (done), AVR2/2-like and AVR4 sequences across the samples to add weight to conclusions about HERB-1 being avirulent. You could also include *Avrblb1*, *avrBlb2*, and *AvrVnt1* in this, as controls. The corresponding *R* genes have not been bred into the cultivated potato, so they represent a contrast in terms of the selection pressures that they refer to in 20th century breeding efforts (i.e., introduction of the demissum *R* genes).

Please respond to these suggestions, including additional data where appropriate, in a revised manuscript.

An expert reviewer on phylogenetic analysis makes the following substantive point:

3) Before running the BEAST analysis, the modern sequences alone are clock tested. I doubt whether these data contain enough signal to detect substantial rate variation, but it is important to test whether there is enough temporal signal in the data to trust the dating analyses.

To do this, it is common to correlate the sampling date against the root-to-tip distance from the ML tree (or NJ tree). This would give a visual indication of the amount of temporal signal in these data (e.g., Harris et al. Science 2010). Second, (and optionally) it is also common to carry out randomisation tests, rerunning the BEAST dating analyses a few times after randomly permuting the sampling dates (see e.g., Firth et al. MBE 2010).

---

## [Author Response]

*Of interest is the apparent increase in ploidy levels in 20th century isolates, compared to HERB-1 and US-1, although it should be stressed that this is based entirely on projected allele frequencies (rather than backed up by other means)*.

Thank you for your interest in the change of ploidy levels in 20th century isolates. To emphasize that the estimation of ploidy levels was based on projected allele frequencies, we changed the sentence in the Discussion as suggested, from “A major genomic difference between the HERB-1 and US‐1 lineages is the shift in ploidy, from diploid to triploid and even tetraploid (Figure 7, and Figure 7–figure supplement 5)”, to “A major genomic difference between the HERB-1 and US‐1 lineages is the shift in ploidy, from diploid to triploid and even tetraploid, which was estimated based on allele frequency from the resequencing alignments (Figure 7, and Figure 7–figure supplement 5)”.

*1) The authors state: “Given that nineteenth century potato strains [surely they mean cultivars?] in North America and Europe did not yet contain resistance genes to control HERB-1, one would expect HERB-1 to contain a full effector gene complement”, as it has not yet been disrupted “…by the selective forces imposed by resistance gene breeding”. What exactly is meant by a “full effector gene complement”? I guess they intend the focus to be on avirulences recognised by the* S. demissum R *genes? These would be AVR1, 2, 3a, 3b, and 4 of those shown in*
Table 2*; the others are not pertinent to their line of logic*.

Thank you for your careful comments. We agree that the expression “full effector gene complement” may lead to a misunderstanding and we modified the text accordingly. As you indicated, our focus is on the genes encoding the avirulence effectors AVR1, AVR2, AVR3a, AVR3b, and AVR4, whose cognate *R* genes were introduced from *S. demissum* through breeding. We rewrote the paragraph on “Effector genes” to better describe the targeted avirulence effectors, particularly those that were the targets of the first breeding efforts.

*In*
Table 2*, the coverage of AVR2 is 100% for the isolate 06_3928A, a “Blue_13” genotype representative. Actually, this genotype lacks AVR2, expressing instead AVR2-like, which suggests that the % coverage they show in this table is lacking the detail to reach their conclusion: “consistent with the expectation that the HERB-1 genotype was avirulent on the first potato cultivars that acquired late blight resistance through breeding”. The focus is on AVR3a, and it is interesting that the KI allele alone exists in the HERB-1 lineage. Did they specifically amplify and sequence this gene from all of the herbarium samples? If their conclusions are based entirely on sequence assemblies (from NGS of ancient DNA), then they need to be backed up by additional experiments. AVR4 has been well documented to be “lost” through mutation that leads to truncated proteins. Is* AVR4 *intact in the historical samples? I am surprised by the apparent absence of* Avr3b. *This is a gene that is conserved in* P. parasitica, *for example. Indeed, they show good coverage in mirabilis and ipomeae in Table 2. Did they specifically check this absence by alternative means (i.e., PCR and sequencing)*?

We appreciate the detailed review of the data and we acknowledge that we did not provide enough details for experts in the field. We now provide details on the alleles of the avirulence effectors we examined in a new table and source data files. In all cases, the alleles recovered are known to be recognized by the corresponding *R* genes and are therefore in their “avirulence” configuration.

We rechecked the coverage of *AVR2 (PITG_22870)* using both alignments to the reference genome and de novo assembly of HERB-1 sequences. The coverage of *AVR2* for the isolate 06_3928A is 81%. There are 15 amino acid differences between the *R2‐recognized AVR2* and unrecognized *AVR2‐like* (FigureS4 in Gilroy et al. New Phytologist 191, 763-76 [2011]). We aligned short reads using BWA software. The BWA cannot align short reads on highly polymorphic regions. This means that we can discriminate *AVR2* and *AVR2‐like* based on the coverage of aligned short reads when the data has enough depth coverage. On the other hand, we could rebuild full length of *AVR2* from HERB-1 reads. We also compared amino acid sequences of HERB-1 with those of the intact *AVR2* and *AVR2‐like*. The deduced HERB-1 *AVR2* sequences were not consistent with those of *AVR2‐like* and instead matched the intact *AVR2* known to have the avirulence activity (25).

We also could not find any frame shift and nonsense mutations in *AVR1, AVR2, AVR3a*, and *AVR4*. We thus conclude that the HERB-1 genotype was presumably avirulent on the first potato cultivars, before these acquired the corresponding late blight resistance genes. We revised the manuscript and added Table 5 showing amino acid differences among *P. infestans* isolates T30-4, HERB-1, DDR7602 (US1-genotype). We also provide a fasta file of HERB-1 effector amino acid sequences to clarify that our conclusion is based on both coverage and amino acid sequences.

Regarding *AVR3b, R3a* and *R3b* are genetically linked, being 0.4 cM apart in the complex *R3* locus (Li, et al. MPMI Vol. 24, No. 10, 2011, pp. 1132–1142). Based on the absence of an *Avr3b* gene in HERB-1, we conclude that initial introgression of the *R3* locus from *S. demissum* was based on resistance conferred by the *R3a* gene. The *R3* phenotype scored during the initial introgression must have been the recognition of *Avr3a* by *R3a*. What would have truly surprising is if both *Avr3a* and *Avr3b* were missing in HERB‐1. We added a paragraph to discuss *Avr3b*.

*2) The reviewers believe that these concerns can be addressed in a straightforward fashion, if you amplify and directly study AVR3a (done), AVR2/2-like and AVR4 sequences across the samples to add weight to conclusions about HERB-1 being avirulent. You could also include Avrblb1, avrBlb2, and AvrVnt1 in this, as controls. The corresponding* R *genes have not been bred into the cultivated potato, so they represent a contrast in terms of the selection pressures that they refer to in 20th century breeding efforts (i.e., introduction of the demissum* R *genes)*.

We agree that PCR amplification and direct sequencing of the *Avr* effector genes of 19th century *P. infestans* would be ideal to complement the described methods and confirm our results based on Illumina short reads. Unfortunately, in general, ancient DNA is heavily fragmentized. In fact, the average length of our 19th century *P. infestans* libraries is rather short (∼70 bp). Hundreds of PCR amplifications would be necessary to reconstruct the effector genes consuming far more DNA extract than what was used for all complete genome reconstructions. Such a project would be extremely challenging and would likely result much more molecular biology effort than the genome wide reconstruction. In fact, PCR approaches have been largely abandoned in the ancient DNA field due to the waste of precious material, a high contamination risk, and their low efficiency (Knapp & Hofreiter, Genes 1, 227-43 [2010]; Stoneking & Krause, *Nat. Rev. Genet* 12, 603-14 [2011]). The whole‐genome sequencing using next-generation sequencing methods approach is therefore currently seen as the most reliable and efficient way to reconstruct ancient genomic regions.

Note also that our analyses were based on both short read alignments and de-novo assembly. These two approaches increase likelihood of accuracy of the deduced sequences of *Avr* genes.

*An expert reviewer on phylogenetic analysis makes the following substantive point*:

*3) Before running the BEAST analysis, the modern sequences alone are clock tested. I doubt whether these data contain enough signal to detect substantial rate variation, but it is important to test whether there is enough temporal signal in the data to trust the dating analyses*.

*To do this, it is common to correlate the sampling date against the root-to-tip distance from the ML tree (or NJ tree). This would give a visual indication of the amount of temporal signal in these data (e.g., Harris et al. Science 2010). Second, (and optionally) it is also common to carry out randomisation tests, rerunning the BEAST dating analyses a few times after randomly permuting the sampling dates (see e.g., Firth et al. MBE 2010)*.

We agree with the reviewer that it should be shown that there is a temporal relationship between the age of a sample and the branch length when we compare HERB-1 and modern strains before we perform a dating analysis. We have therefore added an analysis where we calculate the nucleotide distance between *P. infestans* mtDNAs of the outgroup P17777 to all strains falling into the HERB-1/haplotype Ia/haplotype Ib clade. We grouped the distances in ancient and modern strains and found them to be significantly different. As the reviewer suggested, we further correlated branch length (in nucleotide distance) to the sampling age for each strain and found a strong correlation (r2=0.8). Both analyses were added to the main text and the Materials and methods, and give the dating analysis additional support.

As we have already done three independent Beast runs with 150 million generations each, all of which produced very similar results, we think that further Beast runs are not necessary and would not change our results.